# Mildly Overparametrized Neural Nets can Memorize Training Data Efficiently - Revision

## Abstract

It has been observed (Zhang et al., 2017) that deep neural networks can memorize: they achieve 100% accuracy on training data. Recent theoretical results explained such behavior in highly overparametrized regimes, where the number of neurons in each layer is larger than the number of training samples. In this paper, we show that neural networks can be trained to memorize training data perfectly in a mildly overparametrized regime, where the number of parameters is just a constant factor more than the number of training samples, and the number of neurons is much smaller.

## 1 Introduction

In deep learning, highly non-convex objectives are optimized by simple algorithms such as stochastic gradient descent. There has been many theoretical analysis for the optimization landscape of neural networks(e.g., Brutzkus et al. (2017); Brutzkus & Globerson (2017); Ge et al. (2017b); Wang et al. (2018)), but even very simple two-layer networks have spurious local optima(Safran & Shamir, 2018). In practice, it was observed that neural networks are able to fit the training data perfectly, even when the data/labels are randomly corrupted(Zhang et al., 2017). Recently, a series of work (Du et al. (2019); Allen-Zhu et al. (2019c); Chizat & Bach (2018); Jacot et al. (2018), see more references in Section 1.2) developed a theory of neural tangent kernels (NTK) that explains the success of training neural networks through overparametrization. Several results showed that if the number of neurons at each layer is much larger than the number of training samples, networks of different architectures (multilayer/recurrent) can all fit the training data perfectly.

However, if one considers the number of parameters required for the current theoretical analysis, these networks are highly overparametrized. Consider fully connected networks for example. If a two-layer network has a hidden layer with $r$ neurons, the number of parameters is at least $rd$ where $d$ is the dimension of the input. For deeper networks, if it has two consecutive hidden layers of size $r$, then the number of parameters is at least $r^2$. All of the existing works require the number of neurons $r$ per-layer to be at least the number of training samples $n$ (in fact, most of them require $r$ to be a polynomial of $n$). In these cases, the number of parameters can be at least $nd$ or even $n^2$ for deeper networks –much larger than the number of training samples $n$. Therefore, a natural question is whether neural networks can fit the training data in the mildly overparametrized regime - where the number of parameters is only a constant factor larger than the number of training data. To achieve this, one would want to use a small number of neurons in each layer - $n/d$ for a two-layer network and $\sqrt{n}$ for a three-layer network. Yun et al. (2018) showed such networks have enough capacity to memorize any training data. In this paper we show with polynomial activation functions, simple optimization algorithms are guaranteed to find a solution that memorizes training data.

### 1.1 Our Results

In this paper, we give network architectures (with polynomial activations) such that every layer has width much smaller than the number of training samples $n$, the total number of parameters is linear in $n$, and simple optimization algorithms on such neural networks can fit any training data. We first give a warm-up result that works when the number of training samples is roughly $d^2$ (where $d$ is the input dimension).

**Theorem 1** (Informal). *Suppose there are $n \leq \binom{d+1}{2}$ training samples in general position, there exists a two-layer neural network with quadratic activations, such that the number of neurons in the hidden layer is $2d+2$, the total number of parameters is $O(d^2)$, and perturbed gradient descent can fit the network to any output.*

Here "in general position" will be formalized later as a deterministic condition that is true with probability 1 for random inputs, see Theorem 4 for details.

In this case, the number of hidden neurons is only roughly the square root of the number of training samples, so the weights for these neurons need to be trained carefully in order to fit the data. Our analysis relies on an analysis of optimization landscape - we show that every local minimum for such neural network must also be globally optimal (and has 0 training error). As a result, the algorithm can converge from an arbitrary initialization.

Of course, the result above is limited as the number of training samples cannot be larger than $O(d^2)$. We can extend the result to handle a larger number of training samples:

**Theorem 2** (Informal). *Suppose number of training samples $n \leq d^p$ for some constant $p$, if the training samples are in general position there exists a three-layer neural network with polynomial activations, such that the number of neurons $r$ in each layer is $O_p(\sqrt{n})$, and perturbed gradient descent on the weights of the middle layer can fit the network to any output.*

Here $O_p$ considers $p$ as a constants and hides constant factors that only depend on $p$. We consider "in general position" in the smoothed analysis framework(Spielman & Teng, 2004) - given arbitrary inputs $x_1, x_2, ..., x_n \in \mathbb{R}^d$, fix a perturbation radius $\sqrt{v}$, the actual inputs is $\bar{x}_j = x_j + \tilde{x}_j$ where $\tilde{x}_j \sim N(0, vI)$. The guarantee of training algorithm will depend inverse polynomially on the perturbation $v$ (note that the architecture - in particular the number of neurons - is independent of $v$). The formal result is given in Theorem 5 in Section 4. Later we also give a deterministic condition for the inputs, and prove a slightly weaker result (see Theorem 6).

## 1.2 RELATED WORKS

**Optimization Landscape for Networks without Overparametrization** Many works (Brutzkus & Globerson, 2017; Tian, 2017; Li & Yuan, 2017; Soltanolkotabi, 2017; Zhong et al., 2017; Ge et al., 2017b) analyzed the optimization landscape of 2-layer neural networks. However, these works either work on a single neuron or have very strong assumptions on the input $x$ (such as $x$ is Gaussian). It is also known that even with strong assumptions on input $x$ gradient descent on the standard objective can get stuck in spurious local minima when the network has more than a constant number of neurons (Safran & Shamir, 2018).

**Neural Tangent Kernel** Many results in the framework of neural tangent kernel show that networks with different architecture can all memorize the training data, including two-layer (Du et al., 2019), multi-layer(Du et al., 2018; Allen-Zhu et al., 2019c; Zou & Gu, 2019), recurrent neural network(Allen-Zhu et al., 2019b). However, all of these works require the number of neurons in each layer to be at least quadratic in the number of training samples. Oymak & Soltanolkotabi (2019) improved the number of neurons required for two-layer networks, but their bound is still larger than the number of training samples. There are also more works for NTK on generalization guarantees (e.g., Allen-Zhu et al. (2019a)), fine-grained analysis for specific inputs(Arora et al., 2019b) and empirical performances(Arora et al., 2019c), but they are not directly related to our results.

**Representation Power of Neural Networks** For standard neural networks with ReLU activations, Yun et al. (2018) showed that networks of similar size as Theorem 2 can memorize any training data. Their construction is delicate and it is not clear whether gradient descent can find such a solution.

**Matrix Factorizations and Quadratic Activations** Since the activation function for our two-layer net is quadratic, training of the network is very similar to matrix factorization problem. Many existing works analyzed the optimization landscape and implicit bias for problems related to matrix factorization in various settings(Bhojanapalli et al., 2016; Ge et al., 2016; 2017a; Park et al., 2016; Gunasekar et al., 2017; Li et al., 2018; Arora et al., 2019a). In this line of work, Du & Lee (2018) and Soltanolkotabi et al. (2018) are the most similar to our two-layer result. Du & Lee (2018) showed

how gradient descent can learn a two-layer neural network that represents any positive semidefinite matrix. However positive definite matrices cannot be used to memorize arbitrary data, and our two-layer network can represent an arbitrary matrix. Soltanolkotabi et al. (2018) is very similar to our two-layer result except they require the input to be Gaussian. Extending this condition to our deterministic condition is crucial to our main (3-layer) result.

**Existing works on mildly overparametrization** There are several works on overparametrization that does not fall exactly into the NTK regime. Brutzkus et al. (2017); Wang et al. (2018) works for linear separable setting. However in this setting gradient descent also works even if the network has only a single neuron (no overparametrization), so the results only show that overparametrization does not hurt. Li & Liang (2018) requires an interesting but strong assumption on the input data. Our work is different as we make mild assumptions on the data distribution and the network size is necessary (up to constant factors) to achieve 0 training error without further assumptions.

**Interpolating Methods** Of course, simply memorizing the data may not be useful in machine learning. However, recently several works(Belkin et al., 2018; 2019; Liang et al., 2019; Mei & Montanari, 2019) showed that learning regimes that interpolate/memorize data can also have generalization guarantees. Proving generalization for our architectures is an interesting open problem.

## 2 Preliminaries

In this section, we introduce notations, the two neural network architectures used for Theorem 1 and 2, and the perturbed gradient descent algorithm.

### 2.1 Notations

We use $[n]$ to denote the set $\{1, 2, ..., n\}$. For a vector $x$, we use $\|x\|_2$ to denote its $\ell_2$ norm, and sometimes $\|x\|$ as a shorthand. For a matrix $M$, we use $\|M\|_F$ to denote its Frobenius norm, $\|M\|$ to denote its spectral norm. We will also use $\lambda_i(M)$ and $\sigma_i(M)$ to denote the $i$-th largest eigenvalue and singular value of matrix $M$, and $\lambda_{\min}(M), \sigma_{\min}(M)$ to denote the smallest eigenvalue/singular value.

For the results of three-layer networks, our activation is going to be $x^p$, where $p$ is considered as a small constant. We use $O_p(), \Omega_p()$ to hide factors that only depend on $p$.

For vectors $x, y \in \mathbb{R}^d$, the tensor product is denoted by $(x \otimes x) \in \mathbb{R}^{d^2}$. We use $x^{\otimes p} \in \mathbb{R}^{d^p}$ as a shorthand for $p$-th power of $x$ in terms of tensor product. For two matrices $M, N \in \mathbb{R}^{d_1 \times d_2}$, we use $M \otimes N \in \mathbb{R}^{d_1^2 \times d_2^2}$ denote the Kronecker product of 2 matrices.

### 2.2 Network Architectures

In this section, we introduce the neural net architectures we use. As we discussed, Theorem 1 uses a two-layer network (see Figure 1 (a)) and Theorem 2 uses a three-layer network (see Figure 1 (b)).

**Two-layer Neural Network** For the two-layer neural network, suppose the input samples $x$ are in $\mathbb{R}^d$, the hidden layer has $r$ hidden neurons (for simplicity, we assume $r$ is even, in Theorem 4 we will show that $r = 2d + 2$ is enough). The activation function of the hidden layer is $\sigma(x) = x^2$.

We use $w_i \in \mathbb{R}^d$ to denote the input weight of hidden neuron $i$. These weight vectors are collected as a weight matrix $W = [w_1, w_2, \ldots, w_r] \in \mathbb{R}^{d \times r}$. The output layer has only 1 neuron, and we use $a_i \in \mathbb{R}$ to denote the its input weight from hidden neuron $i$. There is no nonlinearity for the output layer. For simplicity, we fix the parameters $a_i, i \in [r]$ in the way that $a_i = 1$ for all $1 \leq i \leq \frac{r}{2}$ and $a_i = -1$ for all $\frac{r}{2} + 1 \leq i \leq r$. Given $x$ as the input, the output of the neural network is $y = \sum_{i=1}^{r} a_i(w_i^T x)^2$.

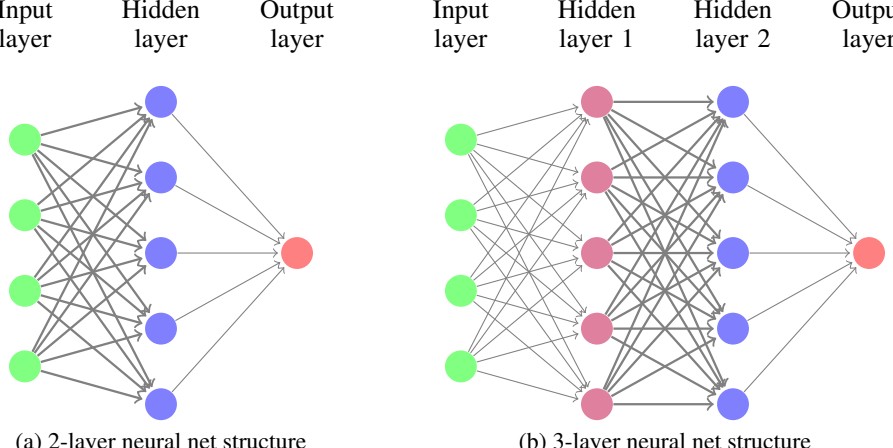

(a) 2-layer neural net structure        (b) 3-layer neural net structure

Figure 1: Neural Network Architectures. The trained layer is in bold face. The activation function after the trained parameters is $x^2$(blue neurons). The activation function before the trained parameters is $x^p$(purple neurons).

If the training samples are $\{(x_j, y_j)\}_{j \leq n}$, we define the empirical risk of the neural network with parameters $W$ to be

$$f(W) = \frac{1}{4n} \sum_{j=1}^{n} \left( \sum_{i=1}^{r} a_i (w_i^T x_j)^2 - y_j \right)^2.$$

**Three-layer neural network** For Theorem 2, we use a more complicated, three-layer neural network. In this network, the first layer has a polynomial activation $\tau(x) = x^p$, and the next two layers are the same as the two-layer network.

We use $R = [r_1, \ldots, r_k]^T \in \mathbb{R}^{k \times d}$ to denote the weight parameter of the first layer. The first hidden layer has $k$ neurons with activation $\tau(x) = x^p$ where $p$ is the parameter in Theorem 2. Given input $x$, the output of the first hidden layer is denoted as $z$, and satisfy $z_i = (r_i^T x_j)^p$. The second hidden layer has $r$ neurons (again we will later show $r = 2k + 2$ is enough). The weight matrix for second layer is denoted as $W = [w_1, \ldots, w_r] \in \mathbb{R}^{k \times r}$ where each $w_i \in \mathbb{R}^k$ is the weight for a neuron in the second hidden layer. The activation for the second hidden layer is $\sigma(x) = x^2$. The third layer has weight $a$ and is initialized the same way as before, where $a_1 = a_2 = \cdots = a_{r/2} = 1$, and $a_{r/2+1} = \cdots = a_r = -1$. The final output $y$ can be computed as $y = \sum_{i=1}^{r} a_i (w_i^T z)^2$.

Given inputs $(x_1, y_1), \ldots, (x_n, y_n)$, suppose $z_i$ is the output of the first hidden layer for $x_i$, the empirical loss is defined as:

$$f(W) = \frac{1}{4n} \sum_{j=1}^{n} \left( \sum_{i=1}^{r} a_i (w_i^T z_j)^2 - y_j \right)^2.$$

Note that only the second-layer weight $W$ is trainable. The first layer with weights $R$ acts like a random feature layer that maps $x_i$'s into a new representation $z_i$'s.

## 2.3 Second order stationary points and perturbed gradient descent

Gradient descent converges to a global optimum of a convex function. However, for non-convex objectives, gradient descent is only guaranteed to converge into a first-order stationary point - a point with 0 gradient, which can be a local/global optimum or a saddle point. Our result requires any algorithm that can find a second-order stationary point - a point with 0 gradient and positive definite Hessian. Many algorithms were known to achieve such guarantee(Ge et al., 2015; Sun et al., 2015; Carmon et al., 2018; Agarwal et al., 2017; Jin et al., 2017a;b). As we require some additional

properties of the algorithm (see Section 3), we will adapt the Perturbed Gradient Descent(PGD, (Jin et al., 2017a)). See Section B for a detailed description of the algorithm. Here we give the guarantee of PGD that we need. The PGD algorithm requires the function and its gradient to be Lipschitz:

**Definition 1** (Smoothness and Hessian Lipschitz). *A differentiable function $f(\cdot)$ is $\ell$-smooth if:*

$$\forall x_1, x_2, \ ||\nabla f(x_1) - \nabla f(x_2)|| \leq \ell ||x_1 - x_2||.$$

*A twice-differentiable function $f(\cdot)$ is $\rho$-Hessian Lipschitz if:*

$$\forall x_1, x_2, \ ||\nabla^2 f(x_1) - \nabla^2 f(x_2)|| \leq \rho ||x_1 - x_2||.$$

Under these assumptions, we will consider an approximation for exact second-order stationary point as follows:

**Definition 2** ($\varepsilon$-second-order stationary point). *For a $\rho$-Hessian Lipschitz function $f(\cdot)$, we say that $x$ is an $\varepsilon$-**second-order stationary point** if: $||\nabla f(x)|| \leq \varepsilon$, and $\lambda_{\min}(\nabla^2 f(x)) \geq -\sqrt{\rho\varepsilon}$.*

Jin et al. (2017a) showed that PGD converges to an $\varepsilon$-second-order stationary point efficiently:

**Theorem 3** (Convergence of PGD (Jin et al. (2017a))). *Assume that $f(\cdot)$ is $\ell$-smooth and $\rho$-Hessian Lipschitz. Then there exists an absolute constant $c_{max}$ such that, for any $\delta > 0, \varepsilon \leq \frac{\ell^2}{\rho}, \Delta_f \geq f(x_0) - f^*$, and constant $c \leq c_{max}$, $PGD(x_0, \ell, \rho, \varepsilon, c, \delta, \Delta_f)$ will output an $\varepsilon$-second-order stationary point with probability $1 - \delta$, and terminate in the following number of iterations:*

$$O\left(\frac{\ell(f(x_0) - f^*)}{\varepsilon^2} \log^4\left(\frac{d\ell\Delta_f}{\varepsilon^2\delta}\right)\right).$$

## 3 WARM-UP: TWO-LAYER NET FOR FITTING SMALL TRAINING SET

In this section, we show how the two-layer neural net in Section 2.2 trained with perturbed gradient descent can fit any small training set (Theorem 1). Our result is based on a characterization of optimization landscape: for small enough $\varepsilon$, every $\varepsilon$-second-order stationary point achieves near-zero training error. We then combine such a result with PGD to show that simple algorithms can always memorize the training data. Detailed proofs are deferred to Section D in the Appendix.

### 3.1 OPTIMIZATION LANDSCAPE OF TWO-LAYER NEURAL NETWORK

Recall that the two-layer network we consider has $r$-hidden units with bottom layer weights $w_1, w_2, ..., w_r$, and the weight for the top layer is set to $a_i = 1$ for $1 \leq i \leq r/2$, and $a_i = -1$ for $r/2 + 1 \leq i \leq r$. For a set of input data $\{(x_1, y_1), (x_2, y_2), ..., (x_n, y_n)\}$, the objective function is defined as $f(W) = \frac{1}{4n} \sum_{j=1}^{n} \left(\sum_{i=1}^{r} a_i(w_i^T x_j)^2 - y_j\right)^2$.

With these definitions, we will that when a point is an approximate second-order stationary point (in fact, we just need it to have an almost positive semidefinite Hessian) it must also have low loss:

**Lemma 1** (Optimization Landscape). *Given training data $\{(x_1, y_1), (x_2, y_2), ..., (x_n, y_n)\}$, Suppose the matrix $X = [x_1^{\otimes 2}, ..., x_n^{\otimes 2}] \in \mathbb{R}^{d^2 \times n}$ has full column rank and the smallest singular value is at least $\sigma$. Also suppose that the number of hidden neurons satisfies $r \geq 2d + 2$. Then if $\lambda_{\min}\nabla^2 f(W) \geq -\varepsilon$, the function value is bounded by $f(W) \leq \frac{nd\varepsilon^2}{4\sigma^2}$.*

Soltanolkotabi et al. (2018) gave a similar characterization of the landscape, except their theorem requires $x_i$'s to be Gaussians.

### 3.2 OPTIMIZING THE TWO-LAYER NEURAL NET

Given the property of the optimization landscape for $f(W)$, it is natural to try to use PGD to find a second-order stationary point. However, this is not enough since the function $f$ does not have bounded smoothness constant and Hessian Lipschitz constant, and without improved analysis PGD is not guaranteed to converge in polynomial time. In order to control the Lipschitz parameters, we

note that these parameters are bounded when the norm of $W$ is bounded (see Lemma 5 in appendix). Therefore we add a small regularizer term to control the norm of $W$. More concretely, we optimize:

$$g(W) = f(W) + \frac{\gamma}{2}||W||_F^2.$$

We want to use this regularizer term to show that: 1. the optimization landscape is preserved: for appropriate $\gamma$, any $\varepsilon$-second-order stationary point of $g(W)$ will still give a small $f(W)$; and 2. During the training process of the 2-layer neural network, the norm of $W$ is bounded, therefore the smoothness and Hessian Lipschitz parameters are bounded. Then, the proof of Theorem 1 just follows from the combination of Theorem 3 of PGD and the result of the geometric property.

The first step is simple as the regularizer only introduces a term $\gamma I$ to the Hessian, which increases all the eigenvalues by $\gamma$. Therefore any $\varepsilon$-second-order stationary point of $g(W)$ will also lead to the fact that $|\lambda_{\min}\nabla^2 f(W)|$ is small, and hence $f(W)$ is small by Lemma 1.

For the second step, note that in order to show the training process using PGD will not escape from the area $\{W : ||W||_F^2 \leq \Gamma\}$ with some $\Gamma$, it suffices to bound the function value $g(W)$ by $\gamma\Gamma/2$, which implies $||W||_F^2 \leq \frac{2}{\gamma}g(W) \leq \Gamma$. To bound the function value we use properties of PGD: for a gradient descent step, since the function is smooth in this region, the function value always decreases; for a perturbation step, the function value can increase, but cannot increase by too much. Using mathematical induction, we can show that the function value of $g$ is smaller than some fixed value(related to the random initialization but not related to time $t$) and will not escape the set $\{W : ||W||_F^2 \leq \Gamma\}$ for appropriate $\Gamma$. Combining these analysis we have the following theorem:

**Theorem 4** (Main theorem for 2-layer NN). *Suppose the matrix $X = [x_1^{\otimes 2}, \ldots, x_n^{\otimes 2}] \in \mathbb{R}^{d^2 \times n}$ has full column rank and the smallest singular value is at least $\sigma$. Also assume that we have $||x_j||_2 \leq B$ and $|y_j| \leq Y$ for all $j \leq n$. We choose our width of neural network $r \geq 2d + 2$ and we choose $\rho = (6B^4\sqrt{2(f(0)+1)})(nd/(\sigma^2\varepsilon))^{1/4}$, $\gamma = (\sigma^2\varepsilon/nd)^{1/2}$, and $\ell = \max\{(3B^4\frac{2(f(0)+1)}{\gamma} + YB^2 + \gamma), 1\}$. Then there exists an absolute constant $c_{max}$ such that, for any $\delta > 0, \Delta \geq f(0) + 1$, and constant $c \leq c_{max}$, $PGD(0, \ell, \rho, \varepsilon, c, \delta, \Delta)$ on $W$ will output an parameter $W^*$ such that with probability $1 - \delta$, $f(W^*) \leq \varepsilon$ when the algorithm terminates in the following number of iterations:*

$$O\left(\frac{B^8\ell(nd)^{5/2}(f(0)+1)^2}{\sigma^5\varepsilon^{5/2}}\log^4\left(\frac{Bnrd\ell\Delta(f(0)+1)}{\varepsilon^2\delta\sigma}\right)\right).$$

## 4 THREE-LAYER NET FOR FITTING LARGER TRAINING SET

In this section, we show how a three-layer neural net can fit a larger training set (Theorem 2). The main limitation of the two-layer architecture in the previous section is that the activation functions are quadratic. Therefore, no matter the number neurons in the hidden layer, the whole network only captures a quadratic function over the input, and cannot fit an arbitrary training set of size much larger than $d^2$. On the other hand, if one replaces the quadratic activation with other functions, it is known that even two-layer neural networks can have bad local minima(Safran & Shamir, 2018).

To address this problem, the three-layer neural net in this section uses the first-layer as a random mapping of the input. The first layer is going to map inputs $x_i$'s into $z_i$'s of dimension $k$ (where $k = \Theta(\sqrt{n})$). If $z_i$'s satisfy the requirements of Theorem 4, then we can use the same arguments as the previous section to show perturbed gradient descent can fit the training data.

We prove our main result in the smoothed analysis setting, which is a popular approach for going beyond worst-case. Given any worst-case input $\{x_1, x_2, ..., x_n\}$, in the smoothed analysis framework, these inputs will first be slightly perturbed before given to the algorithm. More specifically, let $\bar{x}_j = x_j + \tilde{x}_j$, where $\tilde{x}_j \in \mathbb{R}^d$ is a random Gaussian vector following the distribution of $\mathcal{N}(\mathbf{0}, v\mathbf{I})$. Here the amount of perturbation is controlled by the variance $v$. The final running time for our algorithm will depend inverse polynomially on $v$. Note that on the other hand, the network architecture and the number of neurons/parameters in each layer does not depend on $v$.

Let $\{z_1, z_2, ..., z_n\}$ denote the output of the first layer with $(z_j)_i = (r_i^T\bar{x}_j)^p (j = 1, 2, ..., n)$, we first show that $\{z_j\}$'s satisfy the requirement of Theorem 4:

**Lemma 2.** *Suppose $k \leq O_p(d^p)$ and $\binom{k+1}{2} > n$, let $\bar{x}_j = x_j + \tilde{x}_j$ be the perturbed input in the smoothed analysis setting, where $\tilde{x}_j \sim \mathcal{N}(\mathbf{0}, v\mathbf{I})$, let $\{z_1, z_2, ..., z_n\}$ be the output of the first*

*layer on the perturbed input $((z_j)_i = (r_i^T \bar{x}_j)^p)$. Let $Z \in \mathbb{R}^{k^2 \times n}$ be the matrix whose $j$-th column is equal to $z_j^{\otimes 2}$, then with probability at least $1 - \delta$, the smallest singular value of $Z$ is at least $\Omega_p(v^p \delta^{4p} / n^{2p+1/2} k^{4p})$.*

This lemma shows that the output of the first layer ($z_j$'s) satisfies the requirements of Theorem 4. With this lemma, we can prove the main theorem of this section:

**Theorem 5** (Main theorem for 3-layer NN). *Suppose the original inputs satisfy $\|x_j\|_2 \leq 1, |y_j| \leq 1$, inputs $\bar{x}_j = x_j + \tilde{x}_j$ are perturbed by $\tilde{x}_j \sim \mathcal{N}(0, vI)$, with probability $1 - \delta$ over the random initialization, for $k = 2\lceil\sqrt{n}\rceil$, perturbed gradient descent on the second layer weights achieves a loss $f(W^*) \leq \epsilon$ in $O_p(1) \cdot \frac{(n/v)^{O(p)}}{\epsilon^{5/2}} \log^4(n/\epsilon)$ iterations.*

Using different tools, we can also prove a similar result without the smoothed analysis setting:

**Theorem 6.** *Suppose the matrix $X = [x_1^{2p}, ..., x_n^{2p}] \in \mathbb{R}^{d^{2p} \times n}$ has full column rank, and smallest singular value at least $\sigma$. Choose $k = O_p(d^p)$, with high probability perturbed gradient descent on the second layer weights achieves a loss $f(W^*) \leq \epsilon$ in $O_p(1) \cdot \frac{(n)^{O(p)}}{\sigma^5 \epsilon^{5/2}} \log^4(n/\epsilon)$ iterations.*

When the number of samples $n$ is smaller than $d^{2p}/(2p)!$, one can choose $k = O_p(d^p)$, in this regime the result of Theorem 6 is close to Theorem 5. However, if $n$ is just larger, say $n = d^{2p}$, one may need to choose $k = O_p(d^{p+1})$, which gives sub-optimal number of neurons and parameters.

**Proof techniques**  Theorem 5 relies on Theorem 4 – it suffices to prove the two requirements in in Theorem 4: (a) The norm of output from each input sample is bounded with high probability; (b) The matrix formed by the tensor of each output (Matrix $Z$ in Lemma 2) has a high-probability positive smallest singular value lower bound. The proof of (a) is relatively simple as one only needs standard concentration bounds. The condition (b) is stated in Lemma 2.

To prove Lemma 2, we need to show that the vectors $z_j^{\otimes 2}$ are linearly independent (and well-conditioned). We prove that by using a technique called leave-one-out distance, which is widely used in random matrix theory (e.g., in Rudelson & Vershynin (2009)). Roughly speaking, leave-one-out distance requires us to prove that for any fixed $j$, the vector $z_j^{\otimes 2}$ is far from the span of all the other $z_l^{\otimes 2} (l \neq j)$. We in fact show something stronger: for any fixed linear subspace (that is independent of $\tilde{x}_j$) of dimension at most $n - 1$, the distance between $z_j^{\otimes 2}$ and this subspace is large. This is done using anti-concentration inequalities for polynomials. Of course, there are additional challenges in this approach, as the entries of $z_j^{\otimes 2}$ are not independent. We use decoupling techniques to handle the additional dependencies. See details in Appendix E.

## 5 EXPERIMENTS

In this section, we validate our theory using experiments. Detailed parameters of the experiments as well as more result are deferred to Section A in Appendix.

**Small Synthetic Example**  We first run gradient descent on a synthetic data-set, which fits into the setting of Theorem 4. Our training set, including the samples and the labels, are generated from a fixed normalized uniform distribution(random sample from a hypercube and then normalized to have norm 1). As shown in Figure 2, simple gradient descent can already memorize the training set.

**MNIST Experiment**  We also show how our architectures (two-layer and three-layer) can be used to memorize MNIST. For MNIST, we use a squared loss between the network's prediction and the true label (which is an integer in $\{0, 1, ..., 9\}$). For the two-layer experiment, we use the original MNIST dataset, with a small Gaussian perturbation added to the data to make sure the condition in Theorem 4 is satisfied. For the three-layer experiment, we use PCA to project MNIST images to 100 dimensions (so the two-layer architecture will no longer be able to memorize the training set). See Figure 3 for the results. In this part, we use ADAM as the optimizer to improve convergence speed, but given the main result on optimization landscape the algorithm is flexible.

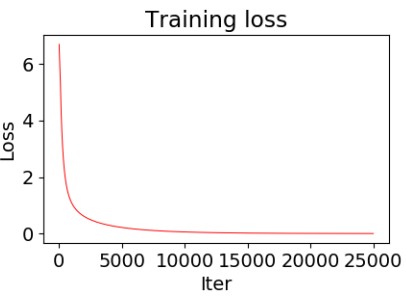

Figure 2: Training loss for random sample experiment

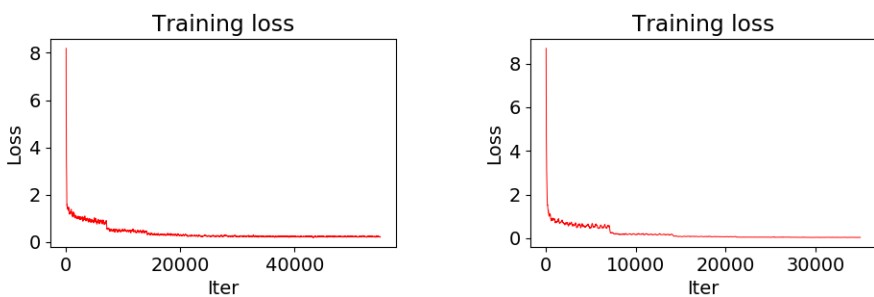

(a) Two-layer network with perturbation on input (b) Three-layer network on top 100 PCA directions

Figure 3: MNIST with original label

**MNIST with random label**    We further test our results on MNIST with random labels to verify that our result does not use any potential structure in the MNIST datasets. The setting is exactly the same as before. As shown in Figure 4, the training loss can also converge.

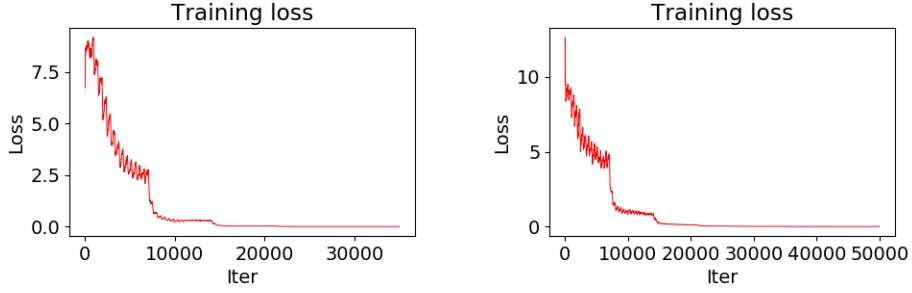

(a) Two-layer network with perturbation on input (b) Three-layer network on top 100 PCA directions

Figure 4: MNIST with random label

## 6    CONCLUSION

In this paper, we showed that even a mildly overparametrized neural network can be trained to memorize the training set efficiently. The number of neurons and parameters in our results are tight (up to constant factors) and matches the bounds in Yun et al. (2018). There are several immediate open problems, including generalizing our result to more standard activation functions and providing generalization guarantees. More importantly, we believe that the mildly overparametrized regime is more realistic and interesting compared to the highly overparametrized regime. We hope this work would be a first step towards understanding the mildly overparametrized regime for deep learning.

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

## A   MORE EXPERIMENTS AND DETAILED EXPERIMENT SETUP

### A.1   EXPERIMENTS SETUP

In this section, we introduce the experiment setup in detail.

**Small Synthetic Example**   We generate the dataset in the following way: We first set up a random matrices $X \in \mathbb{R}^{N \times d}$(samples), where $N$ is the number of samples, $d$ is the input dimension and $Y \in \mathbb{R}^N$(labels). Each entry in $X$ or $Y$ follows a uniform distribution with support $[-1, 1]$. Each entry is independent from others. Then we normalize the dataset $X$ such that each row in $X$ has norm 1, denote the normalized dataset as $\hat{X} = [\hat{x}_1, \ldots, \hat{x}_N]^T$. Then we compute the smallest singular value for the matrix $[\hat{x}_1^{\otimes 2}, \ldots, \hat{x}_N^{\otimes 2}]^T$, and we feed the normalized dataset $\hat{X}$ into the two-layer network(Section 2.2) with $r$ hidden neurons. We select all the parameters as shown in Theorem 4, and plot the function value for $f(\cdot)$.

In our experiment for the small artificial random dataset, we choose $N = 300, d = 100$, and $r = 300$.

**MNIST experiments**   For MNIST, we use a squared loss between the network's prediction and the true label (which is an integer in $\{0, 1, ..., 9\}$).

For the first two-layer network structure, we first normalize the samples in MNIST dataset to have norm 1. Then we set up a two-layer network with quadratic activation with $r = 3000$ hidden neurons (note that although our theory suggests to choose $r = 2d + 2$, having a larger $r$ increases the number of decreasing directions and helps optimization algorithms in practice). For these experiments, we use Adam optimizer(Kingma & Ba, 2014) with batch size 128, initial learning rate 0.003, and decay the learning rate by a factor of 0.3 every 15 epochs (we find that the learning rate-decay is crucial for getting high accuracy).

We run the two-layer network in two settings, one for the original MNIST data, and one for the MNIST data with a small Gaussian noise (0.01 standard deviation per coordinate). The perturbation is added in order for the conditions in Theorem 4 to hold.

For the three-layer network structure, we first normalize the samples in MNIST dataset with norm 1. Then we do the PCA to project it into a 100-dimension subspace. We use $D = [x_1, \ldots, x_n]$ to denote this dataset after PCA. Note that the original 2-layer network may not apply to this setting, since now the matrix $X = [x_1^{\otimes 2}, \ldots, x_n^{\otimes 2}]$ does not have full column rank($60000 > 100^2$). We then add a small Gaussian perturbation to $\tilde{D} \sim \mathcal{N}(0, \sigma_1^2)$ to the sample matrix $D$ and denote the perturbed matrix $\bar{D} = [\bar{x}_1, \ldots, \bar{x}_n]$. We then randomly select a matrix $Q \sim \mathcal{N}(0, \sigma_2^2)^{k \times d}$ and compute the random feature $z_j = (Q\bar{x}_j)^2$, where $(\cdot)^2$ denote the element-wise square. Then we feed this sample into the 2-layer neural network with hidden neuron $d$. Note that this is equivalent to our three-layer network structure in Section 2.2. In our experiments, $k = 750, r = 3000, \sigma_1 = 0.05, \sigma_2 = 0.15$.

**MNIST with random labels**   These experiments have exactly the same set-up as the original MNIST experiments, except that the labels are replaced by a random number in $\{0,1,2,...,9\}$.

### A.2   EXPERIMENT RESULTS

In this section, we give detailed experiment results with bigger plots. For all the training loss graphs, we record the training loss for every 5 iterations. Then for the $i$th recorded loss, we average the recorded loss from $i - 19$th to $i$th and set it as the average loss at $(5i)$th iteration. Then we take the logarithm on the loss and generated the training loss graphs.

**Small Synthetic Example**   As we can see in Figure 5 the loss converges to 0 quickly.

**MNIST experiments with original labels**   First we compare Figure 6 and Figure 7. In Figure 6, we optimize the two-layer architecture with original input/labels. Here the loss decreases to a small value ($\sim 0.1$), but the decrease becomes slower afterwards. This is likely because for the matrix $X$ defined in Theorem 4, some of the directions have very small singular values, which makes it

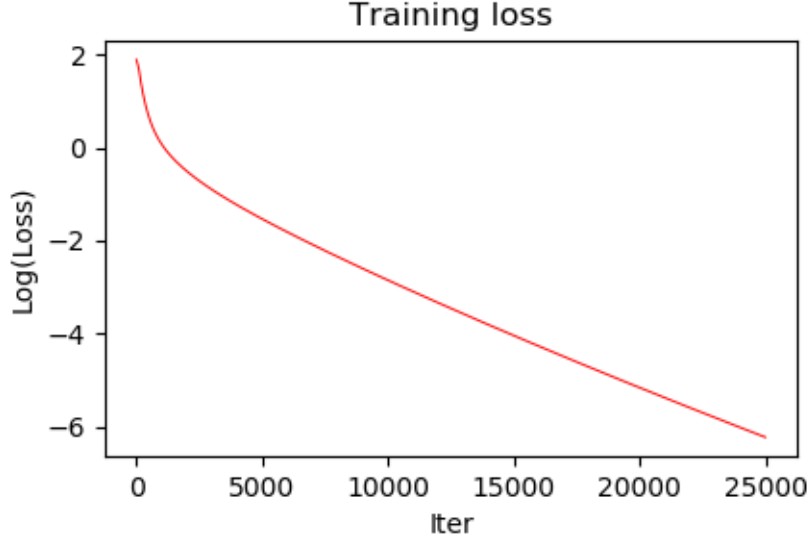

Figure 5: Synthetic Example

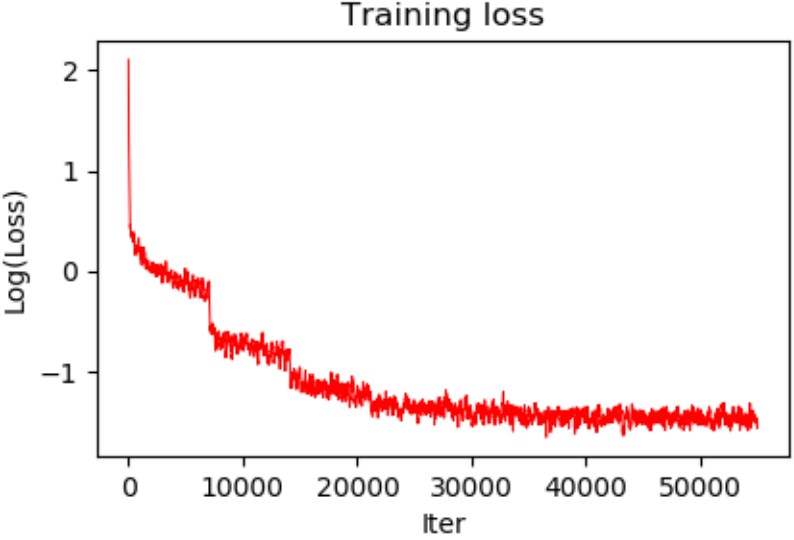

Figure 6: Two-layer network on original MNIST

much harder to correctly optimize for those directions. In Figure 7, after adding the perturbation the smallest singular value of the matrix $X$ becomes better, and as we can see the loss decreases geometrically to a very small value ($< 1e - 5$).

A surprising phenomenon is that even though we offer no generalization guarantees, the network trained as in Figure 6 has an MSE error of 1.21 when tested on test set, which is much better than a random guess (recall the range of labels is 0 to 9). This is likely due to some implicit regularization effect (Gunasekar et al., 2017; Li et al., 2018).

For three-layer networks, in Figure 8 we can see even though we are using only the top 100 PCA directions, the three-layer architecture can still drive the training error to a very low level.

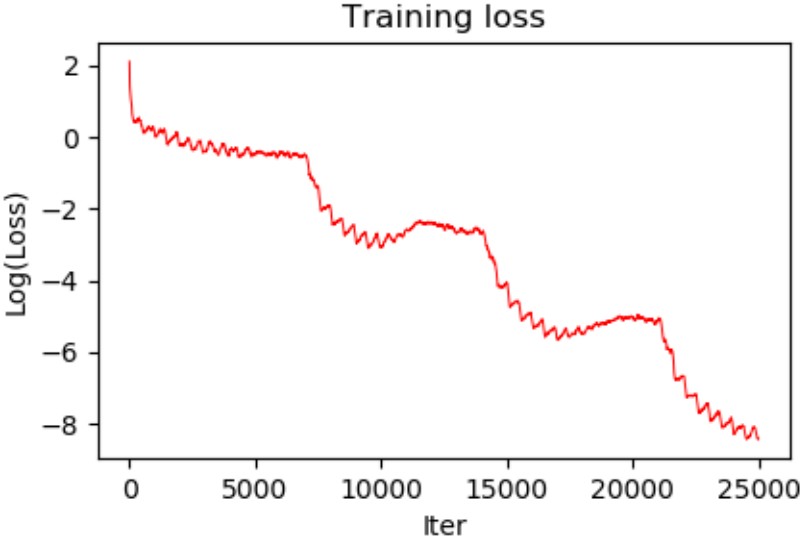

Figure 7: Two-layer network on MNIST, with noise std 0.01

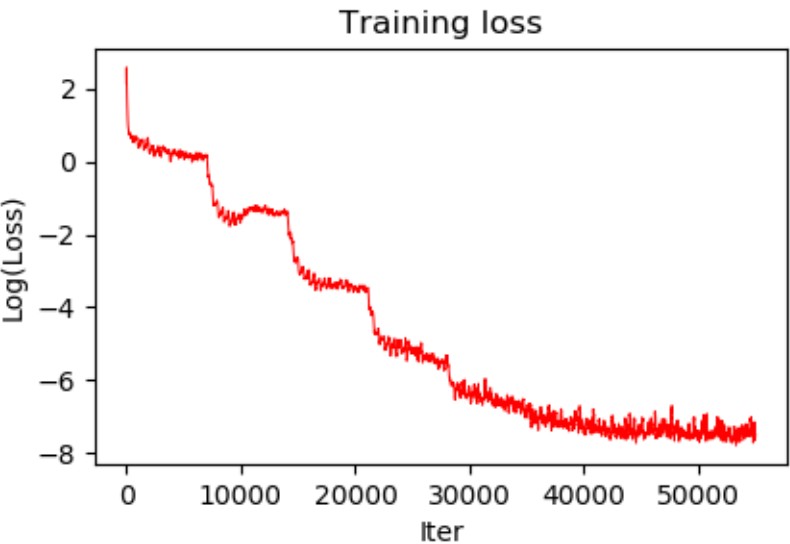

Figure 8: Three-layer network with top 100 PCA directions of MNIST, 0.05 noise per direction

**MNIST with random label**   When we try to fit random labels, the original MNIST input does not work well. We believe this is again because there are many small singular values for the matrix $X$ in Theorem 4, so the data does not have enough effective dimensions fit random labels. The reason that it was still able to fit the original labels to some extent (as in Figure 6) is likely because the original label is correlated with some features of the input, so the original label is less likely to fall into the subspace with smaller singular values. Similar phenomenon was found in Arora et al. (2019b).

Once we add perturbation, for two-layer networks we can fit the random label to very high accuracy, as in Figure 9. The performance for three-layer network in Figure 10 is also similar to Figure 8.

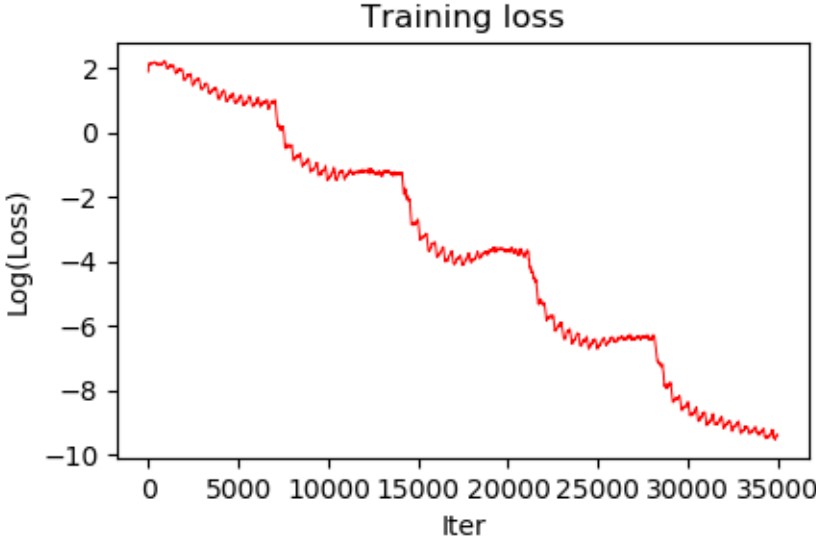

Figure 9: Two-layer network on MNIST, with noise std 0.01, random labels

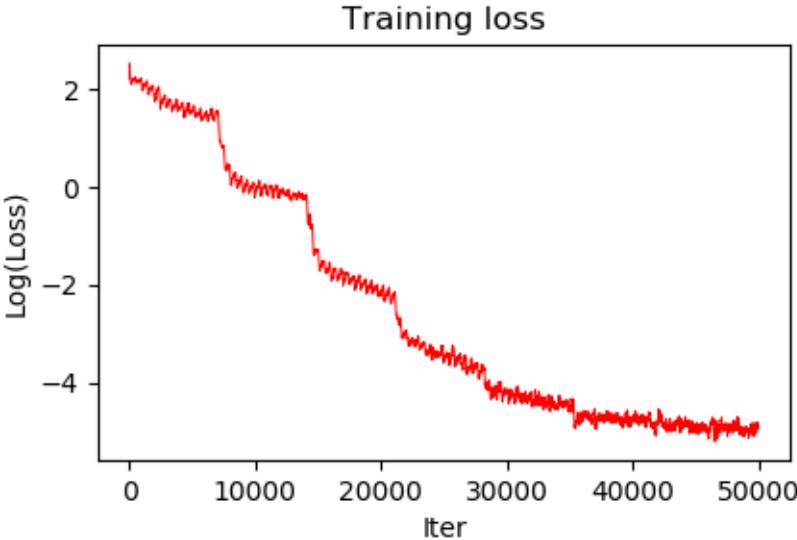

Figure 10: Three-layer network with top 100 PCA directions of MNIST, 0.05 noise per direction, random labels

## B  DETAILED DESCRIPTION OF PERTURBED GRADIENT DESCENT

In this section we give the pseudo-code of the Perturbed Gradient Descent algorithm as in Jin et al. (2017a), see Algorithm 1. The algorithm is quite simple: it just runs the standard gradient descent, except if the loss has not decreased for a long enough time, it adds a perturbation. The perturbation allows the algorithm to escape saddle points. Note that we only use PGD algorithm to find a second-order stationary point. Many other algorithms, including stochastic gradient descent and accelerated gradient descent, are also known to find a second-order stationary point efficiently. All these algorithms can be used for our analysis.

---

**Algorithm 1** Perturbed Gradient Descent

---

**Input:** $x_0, \ell, \rho, \varepsilon, c, \delta, \Delta_f$.

1: $\chi \leftarrow 3\max\left\{\log\left(\frac{d\ell\Delta_f}{c\varepsilon^2\delta}\right), 4\right\}, \eta \leftarrow \frac{c}{\ell}, r \leftarrow \frac{\sqrt{c}\varepsilon}{\chi^2\ell}, g_{\text{thres}} \leftarrow \frac{\sqrt{c}\varepsilon}{\chi^2}, f_{\text{thres}} \leftarrow \frac{c\sqrt{\varepsilon^3}}{\chi^3\sqrt{\rho}}, t_{\text{thres}} \leftarrow \frac{\chi\ell}{c^2\sqrt{\rho\varepsilon}}$

2: $t_{\text{noise}} \leftarrow -t_{\text{thres}} - 1$

3: **for** $t = 0, 1, \ldots$ **do**

4:      **if** $\|\nabla f(x_t)\| \leq g_{\text{thres}}$ and $t - t_{\text{noise}} > t_{\text{thres}}$ **then**

5:          $\tilde{x}_t \leftarrow x_t, t_{\text{noise}} \leftarrow t$

6:          $x_t \leftarrow \tilde{x}_t + \xi_t$, where $\xi_t$ is drawn uniformly from $\mathbb{B}_0(r)$.

7:      **end if**

8:      **if** $t - t_{\text{noise}} = t_{\text{thres}}$ and $f(x_t) - f(\tilde{x}_{t_{\text{noise}}}) > -f_{\text{thres}}$ **then**

9:          **return** $\tilde{x}_{t_{\text{noise}}}$

10:      **end if**

11:      $x_{t+1} \leftarrow x_t - \eta\nabla f(x_t)$

12: **end for**

---

## C   GRADIENT AND HESSIAN OF THE COST FUNCTION

Before we prove any of our main theorems, we first compute the gradient and Hessian of the functions $f(W)$ and $g(W)$. In our training process, we need to compute the gradient of function $g(W)$, and in the analysis for the smoothness and Hessian Lipschitz constants, we need both the gradient and Hessian.

Recall that given the samples and their corresponding labels $\{(x_j, y_j)\}_{j \leq n}$, we define the cost function of the neural network with parameters $W = [w_1, \ldots, w_r] \in \mathbb{R}^{d \times r}$,

$$f(W) = \frac{1}{4n} \sum_{j=1}^{n} \left( \sum_{i=1}^{r} a_i(w_i^T x_j)^2 - y_j \right)^2.$$

Given the above form of the cost function, we can write out the gradient and the hessian with respect to $W$. We have the following gradient,

$$\frac{\partial f(W)}{\partial w_k} = \frac{1}{4n} \sum_{j=1}^{n} 2 \left( \sum_{i=1}^{r} a_i(w_i^T x_j)^2 - y_j \right) \cdot 2a_k(w_k^T x_j)x_j$$

$$= \frac{a_k}{n} \sum_{j=1}^{n} \left( \sum_{i=1}^{r} a_i(w_i^T x_j)^2 - y_j \right) x_j x_j^T w_k.$$

and $\frac{\partial^2 f(W)}{\partial w_{k_1} \partial w_{k_2}} =$

$$\begin{cases} \dfrac{a_{k_1}}{n} \sum_{j=1}^{n} \left( \sum_{i=1}^{r} a_i(w_i^T x_j)^2 - y_j \right) x_j x_j^T + \dfrac{2a_{k_1}a_{k_2}}{n} \sum_{j=1}^{n}(x_j^T w_{k_1})(x_j^T w_{k_2})x_j x_j^T, & \text{if } k_1 = k_2 \\[2em] \dfrac{2a_{k_1}a_{k_2}}{n} \sum_{j=1}^{n}(x_j^T w_{k_1})(x_j^T w_{k_2})x_j x_j^T, & \text{if } k_1 \neq k_2 \end{cases}$$

In the above computation, $\frac{\partial f(W)}{\partial w_k}$ is a column vector and $\frac{\partial^2 f(W)}{\partial w_{k_1} \partial w_{k_2}}$ is a square matrix whose different rows means the derivative to elements in $w_{k_2}$ and different columns represent the derivative to elements in $w_{k_1}$. Then, given the above formula, we can write out the quadratic form of the hessian with respect to the parameters $Z = [z_1, z_2, \ldots, z_r] \in \mathbb{R}^{d \times r}$,

$$\nabla^2 f(W)(Z, Z)$$

$$= \sum_{k=1}^{r} z_k^T \left( \frac{a_k}{n} \sum_{j=1}^{n} \left( \sum_{i=1}^{r} a_i(w_i^T x_j)^2 - y_j \right) x_j x_j^T \right) z_k$$

$$+ \sum_{1 \leq k_1, k_2 \leq r} w_{k_2}^T \left( \frac{2a_{k_1} a_{k_2}}{n} \sum_{j=1}^n (x_j^T w_{k_1})(x_j^T w_{k_2}) x_j x_j^T \right) w_{k_1}$$

$$= \sum_{k=1}^r z_k^T \left( \frac{a_k}{n} \sum_{j=1}^n \left( \sum_{i=1}^r a_i (w_i^T x_j)^2 - y_j \right) x_j x_j^T \right) z_k + \frac{2}{n} \sum_{j=1}^n \left( \sum_{i=1}^r a_i w_i^T x_j x_j^T z_i \right)^2.$$

In order to train this neural network in polynomial time, we need to add a small regularizer to the original ocst function $f(W)$. Let

$$g(W) = f(W) + \frac{\gamma}{2} ||W||_F^2,$$

where $\gamma$ is a constant. Then we can directly get the gradient and the hessian of $g(W)$ from those of $f(W)$. We have

$$\nabla_{w_k} g(W) = \frac{a_k}{n} \sum_{j=1}^n \left( \sum_{i=1}^r a_i (w_i^T x_j)^2 - y_j \right) x_j x_j^T w_k + \gamma w_k$$

$$\nabla_W^2 g(W)(Z, Z) = \sum_{k=1}^r z_k^T \left( \frac{a_k}{n} \sum_{j=1}^n \left( \sum_{i=1}^r a_i (w_i^T x_j)^2 - y_j \right) x_j x_j^T \right) z_k$$

$$+ \frac{2}{n} \sum_{j=1}^n \left( \sum_{i=1}^r a_i w_i^T x_j x_j^T z_i \right)^2 + \gamma ||Z||_F^2.$$

For simplicity, we can use $x_j^T W A W^T x_j - y_j$ to denote $(\sum_{i=1}^r a_i (w_i^T x_j)^2 - y_j$, where $A$ is a diagonal matrix with $A_{ii} = a_i$. Then we have

$$\nabla_W g(W) = \frac{1}{n} \sum_{j=1}^n \left( x_j^T W A W^T x_j - y_j \right) x_j x_j^T W A + \gamma W$$

$$\nabla_W^2 g(W)(Z, Z) = \sum_{k=1}^r z_k^T \left( \frac{a_k}{n} \sum_{j=1}^n \left( x_j^T W A W^T x_j - y_j \right) x_j x_j^T \right) z_k$$

$$+ \frac{2}{n} \sum_{j=1}^n \left( \sum_{i=1}^r a_i w_i^T x_j x_j^T z_i \right)^2 + \gamma ||Z||_F^2.$$

## D    OMITTED PROOFS FOR SECTION 3

In this section, we will give a formal proof of Theorem 4. We will follow the proof sketch in Section 3. First in Section D.1 we prove Lemma 1 which gives the optimization landscape for the two-layer neural network with *large enough* width; then in Section D.2 we will show that the training process on the function with regularization will end in polynomial time.

### D.1    OPTIMIZATION LANDSCAPE OF TWO-LAYER NEURAL NET

In this part we will prove the optimization landscape(Lemma 1) of 2-layer neural network. First we recall Lemma 1.

**Lemma 1** (Optimization Landscape). *Given training data $\{(x_1, y_1), (x_2, y_2), ..., (x_n, y_n)\}$, Suppose the matrix $X = [x_1^{\otimes 2}, ..., x_n^{\otimes 2}] \in \mathbb{R}^{d^2 \times n}$ has full column rank and the smallest singular value is at least $\sigma$. Also suppose that the number of hidden neurons satisfies $r \geq 2d + 2$. Then if $\lambda_{\min} \nabla^2 f(W) \geq -\varepsilon$, the function value is bounded by $f(W) \leq \frac{nd\varepsilon^2}{4\sigma^2}$.*

For simplicity, we will use $\delta_j(W) = \sum_{i=1}^r a_i (w_i^T x_j)^2 - y_j$ to denote the error of the output of the neural network and the label $y_j$. Consider the matrix $M = \frac{1}{n} \sum_{j=1}^n \delta_j x_j x_j^T$. To show that every

$\varepsilon$-second-order stationary point $W$ of $f$ will have small function value $f(W)$, we need the following 2 lemmas.

Generally speaking, the first lemma shows that, when the network is *large enough*, any point with *almost Semi-definite Hessian* will lead to a small spectral norm of matrix $M$.

**Lemma 3.** *When the number of the hidden neurons $r \geq 2d + 2$, we have*

$$\lambda_{\min}\nabla^2 f(W) = -\max_i |\lambda_i(M)|,$$

*where $\lambda_{\min}\nabla^2 f(W)$ denotes the smallest eigenvalue of the matrix $\nabla^2 f(W)$ and $\lambda_i(M)$ denotes the $i$-th eigenvalue of the matrix $M$.*

*Proof.* First note that the equation

$$\lambda_{\min}\nabla^2 f(W) = -\max_i |\lambda_i(M)|$$

is equivalent to

$$\min_{||Z||_F=1} \nabla^2 f(W)(Z, Z) = -\max_{||z||_2=1} |z^T M z|,$$

and we will give a proof of the equivalent form.

First, we show that

$$\min_{||Z||_F=1} \nabla^2 f(W)(Z, Z) \geq -\max_{||z||_2=1} |z^T M z|.$$

Intuitively, this is because $\nabla^2 f(W)$ is the sum of two terms, one of them is always positive semidefinite, and the other term is equivalent to a weighted combination of the matrix $M$ applied to different columns of $Z$.

$$\nabla^2 f(W)(Z, Z)$$
$$= \sum_{k=1}^{r} z_k^T \left( \frac{a_k}{n} \sum_{j=1}^{n} \left( \sum_{i=1}^{r} a_i(w_i^T x_j)^2 - y_j \right) x_j x_j^T \right) z_k + \frac{2}{n} \sum_{j=1}^{n} \left( \sum_{i=1}^{r} a_i w_i^T x_j x_j^T z_i \right)^2$$
$$= \sum_{k=1}^{r} a_k z_k^T M z_k + \frac{2}{n} \sum_{j=1}^{n} \left( \sum_{i=1}^{r} a_i w_i^T x_j x_j^T z_i \right)^2$$
$$\geq \sum_{k=1}^{r} a_k z_k^T M z_k$$
$$\geq -\sum_{k=1}^{r} \max_i |\lambda_i(M)| \cdot ||z_k||_2^2$$
$$= -\max_i |\lambda_i(M)| \cdot ||Z||_F^2.$$

Then we have

$$\min_{||Z||_F=1} \nabla^2 f(W)(Z, Z) \geq \min_{||Z||_F=1} (-\max_i |\lambda_i(M)| \cdot ||Z||_F^2) = -\max_i |\lambda_i M| = -\max_{||z||_2=1} |z^T M z|.$$

For the other side, we show that

$$\min_{||Z||_F=1} \nabla^2 f(W)(Z, Z) \leq -\max_{||z||_2=1} |z^T M z|$$

by showing that there exists $Z, ||Z||_F = 1$ such that $\nabla^2 f(W)(Z, Z) = -\max_{||z||_2=1} |z^T M z|$.

First, let $z_0 = \arg\max_{||z||_2=1} |z^T M z|$. Recall that for simplicity, we assume that $r$ is an even number and $a_i = 1$ for all $i \leq \frac{r}{2}$ and $a_i = -1$ for all $i \geq \frac{r+2}{2}$. If $z_0^T M z_0 < 0$, there exists $u \in \mathbb{R}^r$ such that

    1. $||u||_2 = 1$,

2. $u_i = 0$ for all $i \geq \frac{r+2}{2}$,

3. $\sum_{i=1}^{r} a_i u_i w_i = \mathbf{0}$,

since for constraints 2 and 3, they form a homogeneous linear system, and constraint 2 has $\frac{r}{2}$ equations and constraint 3 has $d$ equations. The total number of the variables is $r$ and we have $r > \frac{r}{2} + d$ since we assume that $r \geq 2d + 2$. Then there must exists $r \neq \mathbf{0}$ that satisfies constraints 2 and 3. Then we normalize that $u$ to have norm $||u||_2 = 1$.

Then, let $Z = z_0 u^T$, we have $||Z||_F^2 = ||z_0||_2^2 \cdot ||u||_2^2 = 1$ and

$$\nabla^2 f(W)(Z, Z) = \sum_{k=1}^{r} a_k z_k^T M z_k + \frac{2}{n} \sum_{j=1}^{n} \left( \sum_{i=1}^{r} a_i w_i^T x_j x_j^T z_i \right)^2$$

$$= \sum_{k=1}^{r} a_k u_k^2 z_0^T M z_0 + \frac{2}{n} \sum_{j=1}^{n} \left( \sum_{i=1}^{r} a_i u_i w_i^T x_j x_j^T z_0 \right)^2$$

$$= z_0^T M z_0 + \frac{2}{n} \sum_{j=1}^{n} \left( \sum_{i=1}^{r} \mathbf{0}^T x_j x_j^T z_0 \right)^2$$

$$= - \max_{||z||_2 = 1} |z^T M z|,$$

where the third equality comes from the fact that $||u||_2^2 = \sum_{i=1}^{r} u_i^2 = 1$, $u_i = 0$ for all $i > \frac{r}{2}$, and $\sum_{i=1}^{r} a_i u_i w_i = \mathbf{0}$. The proof for the case when $z_0^T M z_0 > 0$ is symmetric, except we use the second half of the coordinates (where $a_i = -1$).

$\square$

The next step needs to connect the matrix $M$ and the loss function. In particular, we will show that if the spectral norm of $M$ is small, the loss is also small.

**Lemma 4.** *Suppose the matrix $X = [x_1^{\otimes 2}, \dots, x_n^{\otimes 2}] \in \mathbb{R}^{d^2 \times n}$ has full column rank and the smallest singular value is at least $\sigma$. Then if the spectral norm of the matrix $M = \frac{1}{n} \sum_{j=1}^{n} \delta_j x_j x_j^T$ is upper bounded by $\lambda$, the function value is bounded by*

$$f(W) \leq \frac{nd\lambda^2}{4\sigma^2}.$$

*Proof.* We know that the function value $f(W) = \frac{1}{n} \sum_{j=1}^{n} \delta_j^2 = \frac{1}{n} ||\delta||_2^2$, where $\delta \in \mathbb{R}^n$ is the vector whose $j$-th element is $\delta_j$. Because $X = [x_1^{\otimes 2}, \dots, x_n^{\otimes 2}] \in \mathbb{R}^{d^2 \times n}$ has full column rank and the smallest singular value is at least $\sigma$, we know that for any $v \in \mathbb{R}^n$,

$$||Xv||_2 \geq \sigma_{\min}(X) \cdot ||v||_2 \geq \sigma ||v||_2.$$

Since $M = \frac{1}{n} \sum_{j=1}^{n} \delta_j x_j x_j^T$ is a symmetric matrix, $M$ has $d$ real eigenvalues, and we use $\lambda_1, \dots, \lambda_d$ to denote these eigenvalues. Because we assume that the spectral norm of the matrix $M = \frac{1}{n} \sum_{j=1}^{n} \delta_j x_j x_j^T$ is upper bounded by $\lambda$, which means that $|\lambda_i| \leq \lambda$ for all $1 \leq i \leq d$, and we have

$$||M||_F^2 = \sum_{i=1}^{d} \lambda_i^2 \leq \sum_{i=1}^{d} \lambda^2 = d\lambda^2.$$

Then we can conclude that

$$||M||_F^2 = ||\frac{1}{n} \sum_{j=1}^{n} \delta_j x_j x_j^T||_F^2 = \frac{1}{n^2} ||X\delta||_2^2 \geq \frac{1}{n^2} \sigma^2 ||\delta||_2^2,$$

where the second equation comes from the fact that reordering a matrix to a vector preserves the Frobenius norm.

Then combining the previous argument, we have

$$f(W) = \frac{1}{4n}||\delta||_2^2 \le \frac{n}{4\sigma^2}||M||_F^2 \le \frac{nd\lambda^2}{4\sigma^2}.$$

$\square$

Lemma 1 follows immediately from Lemma 3 and Lemma 4.

## D.2   TRAINING GUARANTEE OF THE TWO-LAYER NEURAL NET

Recall that in order to derive the time complexity for the training procedure, we add a regularizer to the function $f$. More concretely,

$$g(W) = f(W) + \frac{\gamma}{2}||W||_F^2,$$

where $\gamma$ is a constant that we choose in Theorem 4.

To analyze the running time of the PGD algorithm, we first bound the smoothness and Hessian Lipschitz parameters when the Frobenius norm of $W$ is bounded.

**Lemma 5.** *In the set $\{W : ||W||_F^2 \le \Gamma\}$, if we have $||x_j||_2 \le B$ and $|y_j| \le Y$ for all $j \le n$, then*

1. *$\nabla g(W)$ is $(3B^4\Gamma + YB^2 + \gamma)$-smooth.*

2. *$\nabla^2 g(W)$ has $6B^4\Gamma^{\frac{1}{2}}$-Lipschitz Hessian.*

*Proof.* We first figure out the smoothness constant. We have

$$||\nabla g(U) - \nabla g(V)||_F$$

$$=||\frac{1}{n}\sum_{j=1}^{n}\left(x_j^T U A U^T x_j - y_j\right) x_j x_j^T U A + \gamma U - \frac{1}{n}\sum_{j=1}^{n}\left(x_j^T V A V^T x_j - y_j\right) x_j x_j^T V A - \gamma V||_F$$

$$\le||\frac{1}{n}\sum_{j=1}^{n}\left(x_j^T U A U^T x_j - y_j\right) x_j x_j^T U A - \frac{1}{n}\sum_{j=1}^{n}\left(x_j^T V A V^T x_j - y_j\right) x_j x_j^T V A||_F + \gamma||U - V||_F.$$

Then we bound the first term, we have

$$||\frac{1}{n}\sum_{j=1}^{n}\left(x_j^T U A U^T x_j - y_j\right) x_j x_j^T U A - \frac{1}{n}\sum_{j=1}^{n}\left(x_j^T V A V^T x_j - y_j\right) x_j x_j^T V A||_F$$

$$=||\frac{1}{n}\sum_{j=1}^{n}\left(x_j^T U A U^T x_j - y_j\right) x_j x_j^T U A - \frac{1}{n}\sum_{j=1}^{n}\left(x_j^T U A U^T x_j - y_j\right) x_j x_j^T V A$$

$$+ \frac{1}{n}\sum_{j=1}^{n}\left(x_j^T U A U^T x_j - y_j\right) x_j x_j^T V A - \frac{1}{n}\sum_{j=1}^{n}\left(x_j^T U A V^T x_j - y_j\right) x_j x_j^T V A$$

$$+ \frac{1}{n}\sum_{j=1}^{n}\left(x_j^T U A V^T x_j - y_j\right) x_j x_j^T V A - \frac{1}{n}\sum_{j=1}^{n}\left(x_j^T V A V^T x_j - y_j\right) x_j x_j^T V A||_F$$

$$\le||\frac{1}{n}\sum_{j=1}^{n}\left(x_j^T U A U^T x_j - y_j\right) x_j x_j^T U A - \frac{1}{n}\sum_{j=1}^{n}\left(x_j^T U A U^T x_j - y_j\right) x_j x_j^T V A||_F$$

$$+ ||\frac{1}{n}\sum_{j=1}^{n}\left(x_j^T U A U^T x_j - y_j\right) x_j x_j^T V A - \frac{1}{n}\sum_{j=1}^{n}\left(x_j^T U A V^T x_j - y_j\right) x_j x_j^T V A||_F$$

$$+ ||\frac{1}{n}\sum_{j=1}^{n}\left(x_j^T U A V^T x_j - y_j\right) x_j x_j^T V A - \frac{1}{n}\sum_{j=1}^{n}\left(x_j^T V A V^T x_j - y_j\right) x_j x_j^T V A||_F.$$

The first term can be bounded by

$$||\frac{1}{n}\sum_{j=1}^{n}\left(x_j^T U A U^T x_j - y_j\right)x_j x_j^T U A - \frac{1}{n}\sum_{j=1}^{n}\left(x_j^T U A U^T x_j - y_j\right)x_j x_j^T V A||_F$$

$$\leq ||\frac{1}{n}\sum_{j=1}^{n}\left(x_j^T U A U^T x_j\right)x_j x_j^T U A - \frac{1}{n}\sum_{j=1}^{n}\left(x_j^T U A U^T x_j\right)x_j x_j^T V A||_F$$

$$+ ||\frac{1}{n}\sum_{j=1}^{n} y_j x_j x_j^T U A - y_j x_j x_j^T V A||_F$$

$$\leq ||\frac{1}{n}\sum_{j=1}^{n}\left(x_j^T U A U^T x_j\right)x_j x_j^T||_F ||(U-V)A||_F + Y B^2 ||(U-V)A||_F$$

$$\leq B^4 \Gamma ||U-V||_F + Y B^2 ||U-V||_F.$$

Similarly, we can show that

$$||\frac{1}{n}\sum_{j=1}^{n}\left(x_j^T U A U^T x_j - y_j\right)x_j x_j^T V A - \frac{1}{n}\sum_{j=1}^{n}\left(x_j^T U A V^T x_j - y_j\right)x_j x_j^T V A||_F \leq B^4 \Gamma ||U-V||_F,$$

and

$$||\frac{1}{n}\sum_{j=1}^{n}\left(x_j^T U A V^T x_j - y_j\right)x_j x_j^T V A - \frac{1}{n}\sum_{j=1}^{n}\left(x_j^T V A V^T x_j - y_j\right)x_j x_j^T V A||_F \leq B^4 \Gamma ||U-V||_F.$$

Then, we have

$$||\nabla g(U) - \nabla g(V)||_F \leq (3B^4\Gamma + Y B^2 + \gamma)||U-V||_F.$$

Then we bound the Hessian Lipschitz constant. We have

$$|\nabla^2 g(U)(Z,Z) - \nabla^2 g(V)(Z,Z)|$$

$$= |\sum_{k=1}^{r} z_k^T \left(\frac{a_k}{n}\sum_{j=1}^{n}\left(x_j^T U A U^T x_j - y_j\right)x_j x_j^T\right)z_k + \frac{2}{n}\sum_{j=1}^{n}\left(\sum_{i=1}^{r} a_i u_i^T x_j x_j^T z_i\right)^2 + \gamma||Z||_F^2$$

$$- \sum_{k=1}^{r} z_k^T \left(\frac{a_k}{n}\sum_{j=1}^{n}\left(x_j^T V A V^T x_j - y_j\right)x_j x_j^T\right)z_k - \frac{2}{n}\sum_{j=1}^{n}\left(\sum_{i=1}^{r} a_i v_i^T x_j x_j^T z_i\right)^2 - \gamma||Z||_F^2|$$

$$\leq \sum_{k=1}^{r}|z_k^T \left(\frac{a_k}{n}\sum_{j=1}^{n}\left(x_j^T (U A U^T - V A V^T)x_j\right)x_j x_j^T\right)z_k|$$

$$+ \frac{2}{n}\sum_{j=1}^{n}|\left(\sum_{i=1}^{r} a_i u_i^T x_j x_j^T z_i\right)^2 - \left(\sum_{i=1}^{r} a_i v_i^T x_j x_j^T z_i\right)^2|.$$

First we have

$$|z_k^T \left(\frac{a_k}{n}\sum_{j=1}^{n}\left(x_j^T (U A U^T - V A V^T)x_j\right)x_j x_j^T\right)z_k|$$

$$\leq \frac{1}{n}\sum_{j=1}^{n}||\left(x_j^T (U A U^T - V A V^T)x_j\right)x_j x_j^T||_F ||z_k||_2^2$$

$$\leq \frac{1}{n}\sum_{j=1}^{n}||\left(x_j^T (U A U^T - U A V^T + U A V^T - V A V^T)x_j\right)x_j x_j^T||_F ||z_k||_2^2$$

$$\leq 2B^4\Gamma^{\frac{1}{2}}||U-V||_F ||z_k||_2^2,$$

So we can bound the first term by

$$\sum_{k=1}^{r} |z_k^T \left( \frac{a_k}{n} \sum_{j=1}^{n} \left( x_j^T (UAU^T - VAV^T) x_j \right) x_j x_j^T \right) z_k|$$

$$\leq \sum_{k=1}^{r} 2B^4 \Gamma^{\frac{1}{2}} ||U - V||_F ||z_k||_2^2 = 2B^4 \Gamma^{\frac{1}{2}} ||U - V||_F ||Z||_F^2.$$

Then for the second term, note that

$$\sum_{i=1}^{r} a_i u_i^T x_j x_j^T z_i = \langle UA, x_j x_j^T Z \rangle,$$

and we have

$$\frac{2}{n} \sum_{j=1}^{n} | \left( \sum_{i=1}^{r} a_i u_i^T x_j x_j^T z_i \right)^2 - \left( \sum_{i=1}^{r} a_i v_i^T x_j x_j^T z_i \right)^2 |$$

$$= \frac{2}{n} \sum_{j=1}^{n} |\langle UA, x_j x_j^T Z \rangle^2 - \langle VA, x_j x_j^T Z \rangle^2|$$

$$= \frac{2}{n} \sum_{j=1}^{n} |\langle (U - V)A, x_j x_j^T Z \rangle \langle (U + V)A, x_j x_j^T Z \rangle|$$

$$\leq \frac{2}{n} \sum_{j=1}^{n} ||(U - V)A||_F ||x_j x_j^T Z||_F ||(U + V)A||_F ||x_j x_j^T Z||_F$$

$$\leq 4B^4 \Gamma^{\frac{1}{2}} ||U - V||_F ||Z||_F^2,$$

where the first inequality comes from the Cauchy-Schwatz inequality. Combining with the previous computation, we have

$$|\nabla^2 g(U)(Z, Z) - \nabla^2 g(V)(Z, Z)| \leq 6B^4 \Gamma^{\frac{1}{2}} ||U - V||_F ||Z||_F^2.$$

$\square$

We also have the theorem showing the convergence result of Perturbed Gradient Descent(Algorithm 1).

**Theorem 3** (Convergence of PGD (Jin et al. (2017a))). *Assume that $f(\cdot)$ is $\ell$-smooth and $\rho$-Hessian Lipschitz. Then there exists an absolute constant $c_{max}$ such that, for any $\delta > 0, \varepsilon \leq \frac{\ell^2}{\rho}, \Delta_f \geq f(x_0) - f^*$, and constant $c \leq c_{max}$, $PGD(x_0, \ell, \rho, \varepsilon, c, \delta, \Delta_f)$ will output an $\varepsilon$-second-order stationary point with probability $1 - \delta$, and terminate in the following number of iterations:*

$$O \left( \frac{\ell(f(x_0) - f^*)}{\varepsilon^2} \log^4 \left( \frac{d\ell \Delta_f}{\varepsilon^2 \delta} \right) \right).$$

Then based on the convergence result in Jin et al. (2017a) and the previous lemmas, we have the following main theorem for 2-layer neural network with quadratic activation.

**Theorem 4** (Main theorem for 2-layer NN). *Suppose the matrix $X = [x_1^{\otimes 2}, \ldots, x_n^{\otimes 2}] \in \mathbb{R}^{d^2 \times n}$ has full column rank and the smallest singular value is at least $\sigma$. Also assume that we have $||x_j||_2 \leq B$ and $|y_j| \leq Y$ for all $j \leq n$. We choose our width of neural network $r \geq 2d + 2$ and we choose $\rho = (6B^4 \sqrt{2(f(0) + 1)}) (nd/(\sigma^2 \varepsilon))^{1/4}$, $\gamma = (\sigma^2 \varepsilon/nd)^{1/2}$, and $\ell = \max\{(3B^4 \frac{2(f(0)+1)}{\gamma} + YB^2 + \gamma), 1\}$. Then there exists an absolute constant $c_{max}$ such that, for any $\delta > 0, \Delta \geq f(0) + 1$, and constant $c \leq c_{max}$, $PGD(0, \ell, \rho, \varepsilon, c, \delta, \Delta)$ on $W$ will output an parameter $W^*$ such that with probability $1 - \delta$, $f(W^*) \leq \varepsilon$ when the algorithm terminates in the following number of iterations:*

$$O \left( \frac{B^8 \ell (nd)^{5/2} (f(0) + 1)^2}{\sigma^5 \varepsilon^{5/2}} \log^4 \left( \frac{Bnrd\ell \Delta(f(0) + 1)}{\varepsilon^2 \delta \sigma} \right) \right).$$

*Proof of Theorem 4.* We first show that during the training process, if the constant $c \leq 1$, the objective function value satisfies

$$g(W_t) \leq g(W_{\text{ins}}) + \frac{3c\varepsilon^2}{2\chi^4},$$

where we choose the smoothness constant $\ell \geq 1$ to be the smoothness for the region $g(W) \leq g(W_{\text{ins}}) + \frac{3c\varepsilon^2}{2\chi^4}$.

In the PGD algorithm (Algorithm 1), we say a point is in a perturbation phase, if $t - t_{noise} < t_{thres}$. A point $x_t$ is the beginning of a perturbation phase if it reaches line 5 of Algorithm 1 and a perturbation is added to it.

We use induction to show that the following properties hold.

1. If time $t$ is not in the perturbation phase, then $g(W_t) \leq g(W_{\text{ins}})$.

2. If time $t$ is in a perturbation phase, then $g(W_t) \leq g(W_{\text{ins}}) + \frac{3c\varepsilon^2}{2\chi^4\ell}$. Moreover, if $t$ is the beginning of a perturbation phase, then $g(\tilde{W}_t) \leq g(W_{\text{ins}})$.

First we show that at time $t = 0$, the property holds. If $t = 0$ is not the beginning of a perturbation phase, then the inequality holds trivially by initialization. If $t = 0$ is the beginning of a perturbation phase, then we know that $g(\tilde{W}_0) = g(W_{\text{ins}})$ from the definition of the algorithm, then

$$\begin{aligned}
g(W_0) =& g(\tilde{W}_0 + \xi_0) \qquad\qquad\qquad\qquad\qquad\qquad\qquad (1)\\
\leq& g(\tilde{W}_0) + ||\xi_0||_F ||\nabla g(\tilde{W}_0)||_F + \frac{\ell}{2}||\nabla g(\tilde{W}_0)||_F^2\\
\leq& g(\tilde{W}_0) + r \cdot g_{\text{thres}} + \frac{\ell}{2}r^2\\
\leq& g(\tilde{W}_0) + \frac{\sqrt{c}\varepsilon}{\chi^2\ell} \cdot \frac{\sqrt{c}\varepsilon}{\chi^2} + \frac{\ell}{2}\frac{\sqrt{c}\varepsilon}{\chi^2\ell} \cdot \frac{\sqrt{c}\varepsilon}{\chi^2\ell}\\
=& g(W_{\text{ins}}) + \frac{3c\varepsilon^2}{2\chi^4\ell}.
\end{aligned}$$

Now we do the induction: assuming the two properties hold for time $t$, we will show that they also hold at time $t + 1$. We break the proof into 3 cases:

**Case 1**: $t + 1$ is not in a perturbation phase. In this case, the algorithm does not add a perturbation on $W_{t+1}$, and we have

$$\begin{aligned}
g(W_{t+1}) =& g(W_t - \eta\nabla g(W_t)) \qquad\qquad\qquad\qquad\qquad\qquad (2)\\
\leq& g(W_t) - \langle \eta\nabla g(W_t), \nabla g(W_t)\rangle + \frac{\ell}{2}||\eta\nabla g(W_t)||_F^2\\
\leq& g(W_t) - \frac{\eta}{2}||\nabla g(W_t)||_F^2||\\
\leq& g(W_t).
\end{aligned}$$

If $t$ is not in a perturbation phase, then from the induction hypothesis, we have

$$g(W_{t+1}) \leq g(W_t) \leq g(W_{\text{ins}}),$$

otherwise if $t$ is in a perturbation phase, since $t + 1$ is not in a perturbation phase, $t$ must be at the end of the phase. By design of the algorithm we have:

$$g(W_{t+1}) \leq g(W_t) \leq g(\tilde{W}_{t_{\text{noise}}}) - f_{\text{thres}} \leq g(W_{\text{ins}}).$$

**Case 2**: $t + 1$ is in a perturbation phase, but not at the beginning. Using the same reasoning as equation 2, we know

$$g(W_{t+1}) \leq g(W_t) \leq g(W_{\text{ins}}).$$

**Case 3**: $t + 1$ is at the beginning of a perturbation phase. First we know that

$$g(W_t) \leq g(W_{\text{ins}}),$$

since $t$ is either not in a perturbation phase of at the end of a perturbation phase, then we have $g(\tilde{W}_{t+1}) \leq g(W_{\text{ins}})$. Same as the computation in equation 1, we have

$$g(W_{t+1}) \leq g(W_{\text{ins}}) + \frac{3c\varepsilon^2}{2\chi^4 \ell}.$$

This finishes the induction.

Since we choose $\ell \geq 1$, we can choose the other parameters such that $g(W_{t+1}) \leq g(W_{\text{ins}}) + \frac{3c\varepsilon^2}{2\chi^4} \leq g(W_{\text{ins}}) + 1$. Then since

$$g(W) = f(W) + \frac{\gamma}{2} ||W||_F^2,$$

we know that during the training process, we have $||W||_F^2 \leq \frac{2(g(W_{\text{ins}})+1)}{\gamma}$. Since we train from $W_{\text{ins}} = 0$, we have $||W||_F^2 \leq \frac{2(f(0)+1)}{\gamma}$. From Lemma 5, we know that

1. $\nabla g(W)$ is $(3B^4 \frac{2(f(0)+1)}{\gamma} + YB^2 + \gamma)$-smooth.

2. $\nabla^2 g(W)$ has $6B^4 \sqrt{\frac{2(f(0)+1)}{\gamma}}$-Lipschitz Hessian.

As we choose $\gamma = (6B^4 \sqrt{2(f(0)+1)})^{2/5} \cdot \varepsilon^{2/5}$, we know that $\rho = (6B^4 \sqrt{2(f(0)+1)})^{4/5} \cdot \varepsilon^{-1/5}$ is an upper bound on the Lipschitz Hessian constant.

When PGD stops, we know that

$$\lambda_{\min}(\nabla^2 g(W)) \geq -\sqrt{\rho\varepsilon} = -(6B^4 \sqrt{2(f(0)+1)})^{2/5} \cdot \varepsilon^{2/5},$$

and we have

$$\lambda_{\min}(\nabla^2 f(W)) \geq \lambda_{\min}(\nabla^2 g(W)) - \gamma \geq -2(6B^4 \sqrt{2(f(0)+1)})^{2/5} \cdot \varepsilon^{2/5}.$$

From Lemma 3, we know that the spectral norm of matrix $M$ is bounded by $2(6B^4 \sqrt{2(f(0)+1)})^{2/5} \cdot \varepsilon^{2/5}$, and from Lemma 4, we know that

$$f(W) \leq \frac{nd \cdot 4(6B^4 \sqrt{2(f(0)+1)})^{4/5} \cdot \varepsilon^{4/5}}{4\sigma^2} = \frac{nd \cdot (6B^4 \sqrt{2(f(0)+1)})^{4/5} \cdot \varepsilon^{4/5}}{\sigma^2}.$$

The running time follows directly from the convergence theorem of Perturbed Gradient Descent(Theorem 3) and the previous argument that the training trajectory will not escape from the set $\{W : ||W||_F^2 \leq \frac{2(g(W_{\text{ins}})+1)}{\gamma}\}$.

Then, in order to get the error to be smaller than $\varepsilon$, we choose

$$\varepsilon' = \left(\frac{\sigma^2 \varepsilon}{nd}\right)^{5/4} \frac{1}{6B^4 \sqrt{2(f(0)+1)}},$$

and the total running time should be

$$O\left(\frac{B^8 \ell(nd)^{5/2}(f(0)+1)^2}{\sigma^5 \varepsilon^{5/2}} \log^4 \left(\frac{Bnrd\ell\Delta(f(0)+1)}{\varepsilon^2 \delta\sigma}\right)\right).$$

Besides, our parameter $\rho$ and $\gamma$ is chosen to be

$$\rho = (6B^4 \sqrt{2(f(0)+1)})^{4/5} \cdot \varepsilon'^{-1/5} = (6B^4 \sqrt{2(f(0)+1)}) \left(\frac{nd}{\sigma^2 \varepsilon}\right)^{\frac{1}{4}},$$

and

$$\gamma = (6B^4 \sqrt{2(f(0)+1)})^{2/5} \cdot \varepsilon'^{2/5} = \left(\frac{\sigma^2 \varepsilon}{nd}\right)^{\frac{1}{2}}.$$

$\square$

# E   OMITTED PROOFS IN SECTION 4

In this section, we give the proof of the main results of our three-layer neural network(Theorem 5 and 6). Our proof mostly uses leave-one-out distance to bound the smallest singular value of the relevant matrices, which is a common approach in random matrix theory (e.g., in ). However, the matrices we are interested in involves high order tensor powers that have many correlated entries, so we need to rely on tools such as anti-concentration for polynomials in order to bound the leave-one-out distance.

First in Section E.1, we introduce some more notations and definitions, and present some well-known results that will help us present the proofs. In Section E.2, we proof Theorem 5 which focus on the smoothed analysis setting. Finally in Section E.3 we prove Theorem 6 where we can give a deterministic condition for the input.

## E.1   PRELIMINARIES

**Representations of symmetric tensors**   Throughout this section, we use $T_d^p$ to denote the space of $p$-th order tensors on $d$ dimensions. That is, $T_d^p = (\mathbb{R}^d)^{\otimes p}$. A tensor $T \in T_d^p$ is symmetric if $T(i_1, i_2, ..., i_p) = T(i_{\pi(1)}, i_{\pi(2)}, ..., i_{\pi(p)})$ for every permutation $\pi$ from $[p] \to [p]$. We use $X_d^p$ to denote the space of all symmetric tensors in $T_d^p$. The dimension of $X_d^p$ is $D_d^p = \binom{p+d-1}{p}$.

Let $\bar{X}_d^p = \left\{ x \in X_d^p \middle| \|x\|_2 = 1 \right\}$ be the set of unit tensors in $X_d^p$ (as a sub-metric space of $T_d^p$). For $\mathbb{R}^d$, let $\{e_i | i = 1, 2 \cdots d\}$ be its standard orthonormal basis. For simplicity of notation we use $S_p$ to denote the group of bijections (permutations) $[p] \to [p]$, and $I_d^p$ to denote the set of integer indices $I_d^p = \{(i_1, i_2 \cdots i_d) \in \mathbb{N}^d | \sum_{j=1}^{d} i_j = p\}$. We can make $X_d^p$ isomorphic (as a vector space over $\mathbb{R}$) to Euclidean space $\mathbb{R}^{I_d^p}$ ($|I_d^p| = D_d^p$) by choosing a basis $\{s_{(i_1, i_2 \cdots i_d) \in I_d^p} = \frac{1}{\prod_{j=1}^{d} i_j!} \sum_{\sigma \in S_p} e_{j_{\sigma(1)}} \otimes e_{j_{\sigma(2)}} \otimes \cdots \otimes e_{j_{\sigma(p)}} | (j_1 \circ j_2 \circ \cdots \circ j_p) = (1^{(i_1)} \circ 2^{(i_2)} \circ \cdots \circ d^{(i_d)})\}$ where $(1^{(i_1)} \circ 2^{(i_2)} \circ \cdots \circ d^{(i_d)})$ means a length $p$ string with $i_1$ 1's, $i_2$ 2's and so on, and let the isomorphism be $\phi_d^p$. We call the image of a symmetric tensor through $\phi_d^p$ its *reduced vectorized form*, and we can define a new norm on $X_d^p$ with $\|x\|_{\mathrm{rv}} = \|\phi_d^p(x)\|_2$.

Given the definition of *reduced vectorized form* and the norm $\| \cdot \|_{\mathrm{rv}}$, we have the following lemma that bridges between the norm $\| \cdot \|_{\mathrm{rv}}$ and the original 2-norm.

**Lemma 6.** *For any $x \in X_n^p$,*

$$\|x\|_{rv} \geq \frac{1}{\sqrt{p!}} \|x\|_2.$$

*Proof.* We can expand $x$ as $x = \sum_{i \in I_n^p} x_i s_i$. Then $\|x\|_{\mathrm{rv}} = \sqrt{\sum_{i \in I_n^p} x_i^2}$ and $\|x\|_2 = \sqrt{\sum_{i \in I_n^p} x_i^2 \|s_i\|_2^2}$ as $\{s_i\}$ are orthogonal. Notice that for $i = (i_1, i_2 \cdots i_n)$, $\|s_i\|_2^2 = \frac{p!}{\prod_{j=1}^{n} i_j!} \leq p!$, and therefore

$$\|x\|_2 \leq \sqrt{\sum_{i \in I_n^p} x_i^2 p!} = \sqrt{p!} \|x\|_{\mathrm{rv}}.$$

$\square$

$\varepsilon$**-net**   Part of our proof uses $\varepsilon$-nets to do a covering argument. Here we give its definition.

**Definition 3** ($\varepsilon$-Net)**.** *Given a metric space $(X, d)$. A finite set $N \subseteq \mathcal{P}$ is called an $\varepsilon$-net for $\mathcal{P} \subset X$ if for every $\boldsymbol{x} \in \mathcal{P}$, there exists $\pi(\boldsymbol{x}) \in N$ such that $d(\boldsymbol{x}, \pi(\boldsymbol{x})) \leq \varepsilon$. The smallest cardinality of an $\varepsilon$-net for $\mathcal{P}$ is called the covering number: $\mathcal{N}(\mathcal{P}, \varepsilon) = \inf\{|N| : N \text{ is an } \varepsilon\text{-net of } \mathcal{P}\}$.*

Then we introduce give an upper bound on the size of $\varepsilon$-net of a set $K \subseteq \mathcal{R}^d$. First, we need the definition of Minkowski sum

**Definition 4** (Minkowski sum). *Let $A, B \subseteq \mathcal{R}^d$ be 2 subsets of $\mathcal{R}^d$, then the Minkowski sum is defined as*

$$A + B := \{a + b : a \in A, b \in B\}.$$

Then the covering number can be bounded by a volume argument. This is well-known, and the proof can be found in Vershynin (2018)(Proposition 4.2.12 in Vershynin (2018)).

**Proposition 1** (Covering number). *Given a set $K \subseteq \mathcal{R}^d$ and the corresponding metric $d(x, y) := \|x - y\|_2$. Suppose that $\varepsilon > 0$, and then we have*

$$\mathcal{N}(K, \varepsilon) \leq \frac{|K + \mathbb{B}_2^d(\varepsilon/2)|}{|\mathbb{B}_2^d(\varepsilon/2)|},$$

*where $|\cdot|$ denote the volume of the set.*

Then with the help of the previous proposition, we can now bound the covering number of symmetric tensors with unit length.

**Lemma 7** (Covering number of $\bar{X}_d^p$). *There exists an $\varepsilon$-net of $\bar{X}_d^p$ with size $O\left(\left(1 + \frac{2\sqrt{p!}}{\varepsilon}\right)^{D_d^p}\right)$, i.e.*

$$\mathcal{N}(\bar{X}_d^p, \varepsilon) \leq O\left(\left(1 + \frac{2\sqrt{p!}}{\varepsilon}\right)^{D_d^p}\right).$$

*Proof.* Recall that $\phi_d^p(\cdot) : \mathbb{R}^{d^p} \to \mathbb{R}^{D_d^p}$ is an bijection between the symmetric tensors in $\mathbb{R}^{d^p}$ and a vector in $\mathbb{R}^{D_d^p}$. We first show that an $\frac{\varepsilon}{\sqrt{p!}}$-net for the image $\phi_d^p(\bar{X}_d^p)$ implies an $\varepsilon$-net for the unit symmetric tensor $\bar{X}_d^p$.

Suppose that the $\frac{\varepsilon}{\sqrt{p!}}$-net for the image $\phi_d^p(\bar{X}_d^p)$ is denoted as $N \subset \phi_d^p(\bar{X}_d^p)$, and for any $x \in \phi_d^p(\bar{X}_d^p)$, there exists $\pi(x) \in N$ such that $\|\pi(x) - x\|_2 \leq \frac{\varepsilon}{\sqrt{p!}}$. Then we know that $(\phi_d^p)^{-1}(N)$ is an $\varepsilon$-net for the unit symmetric tensors $\bar{X}_d^p$, because for any $x' \in \bar{X}_d^p$, we have

$$\begin{aligned}
\|x' - (\phi_d^p)^{-1}(\pi(\phi_d^p(x')))\|_2 &\leq \sqrt{p!}\|\phi_d^p(x') - \pi(\phi_d^p(x'))\|_2 \\
&\leq \sqrt{p!} \cdot \frac{\varepsilon}{\sqrt{p!}} \\
&= \varepsilon,
\end{aligned}$$

where the first inequality comes from Lemma 6.

Next, we bound the covering number for the set $\phi_d^p(\bar{X}_d^p)$. First note that the set satisfies $\phi_d^p(\bar{X}_d^p) \subset \mathbb{R}^{D_d^p}$, and from Proposition 1, we have

$$\begin{aligned}
\mathcal{N}\left(\phi_d^p(\bar{X}_d^p), \frac{\varepsilon}{\sqrt{p!}}\right) &\leq \frac{\left|\phi_d^p(\bar{X}_d^p) + \mathbb{B}_2^{D_d^p}(\frac{\varepsilon}{2\sqrt{p!}})\right|}{\left|\mathbb{B}_2^{D_d^p}(\frac{\varepsilon}{2\sqrt{p!}})\right|} \\
&\leq \frac{\left|\mathbb{B}_2^{D_d^p}(1) + \mathbb{B}_2^{D_d^p}(\frac{\varepsilon}{2\sqrt{p!}})\right|}{\left|\mathbb{B}_2^{D_d^p}(\frac{\varepsilon}{2\sqrt{p!}})\right|} \\
&= \left(1 + \frac{2\sqrt{p!}}{\varepsilon}\right)^{D_d^p},
\end{aligned}$$

where the first inequality comes from Proposition 1 and the second inequality comes from the fact that $\|\phi_d^p(x)\|_2 \leq \|x\|_2$. $\qquad \square$

**Leave-one-out Distance**   Another main ingredient in our proof is *Leave-one-out distance*. This is a notion that is closely related to the smallest singular value, but usually much easier to compute and bound. It has been widely used in random matrix theory, for example in Rudelson & Vershynin (2009).

**Definition 5** (Leave-one-out distance). *For a set of vectors $V = \{v_1, v_2 \cdots v_n\}$, their leave-one-out distance is defined as*

$$l(V) = \min_{1 \leq i \leq n} \inf_{a_1, a_2 \cdots a_n \in R} \|v_i - \sum_{j \neq i} a_j v_j\|_2.$$

*For a matrix $M$, its leave-one-out distance $l(M)$ is the leave-one-out distance of its columns.*

The leave-one-out distance is connected with the smallest singular value by the following lemma:

**Lemma 8** (Leave-one-out distance and smallest singular value). *For a matrix $M \in R^{m \times n}$ with $m \geq n$, let $l(M)$ denote the leave-one-out distance for the columns of $M$, and $\sigma_{\min}(M)$ denote the smallest singular value of $M$, then*

$$\frac{l(M)}{\sqrt{n}} \leq \sigma_{\min}(M) \leq l(M).$$

We give the proof for completeness.

*Proof.* For any $x \in \mathbb{R}^n \backslash \{0\}$, let $r(x) = \underset{i \in [n]}{\operatorname{argmax}} |x_i|$, then $|x_{r(x)}| > 0$ for $x \neq 0$.

Because $l(M) = \min_{i \in [n]} \inf_{x \in \mathbb{R}^n, x_i = 1} \|Mx\|_2$, we have

$$\sigma_{\min}(M) = \inf_{x \in R^n \backslash 0} \frac{\|Mx\|_2}{\|x\|_2}$$

$$= \min_{i \in [n]} \inf_{x \in R^n \backslash 0, r(x) = i} \frac{\|M \frac{x}{x_i}\|_2}{\|\frac{x}{x_i}\|_2}$$

$$= \min_{i \in [n]} \inf_{x' \in R^n \backslash 0, x'_i = 1} \frac{\|Mx'\|_2}{\|x'\|_2}.$$

Because of the equations $\|x'\|_2 \geq |x'_i| = 1$ and $\|x'\|_2 = \sqrt{\sum_{j \in [n]} x_j^2} \leq \sqrt{n}|x'_i| = \sqrt{n}$, we have $\frac{l(M)}{\sqrt{n}} \leq \sigma_{\min}(M) \leq l(M)$. $\square$

**Anti-concentration**   To make use of the random Gaussian noise added in the smoothed analysis setting, we rely on the following anti-concentration result by Carbery & Wright (2001):

**Proposition 2** (Anti-concentration (Carbery & Wright (2001))). *For a multivariate polynomial $f(x) = f(x_1, x_2 \cdots x_n)$ of degree $p$, let $x \sim \mathcal{N}(0, 1)^n$ follows the standard normal distribution, and $\operatorname{Var}[f] \geq 1$, then for any $t \in \mathbb{R}$ and $\varepsilon > 0$,*

$$\Pr_x[|f(x) - t| \leq \varepsilon] \leq O(p)\varepsilon^{1/p} \tag{3}$$

**Gaussian moments**   To apply the anti-concentration result, we need to give lower bound of the variance of a polynomial when the variables follow standard Gaussian distribution $\mathcal{N}(0, 1)$. Next, we will show some definitions, propositions, and lemmas that will help us to give lower bound for variance of polynomials.

**Proposition 3** (Gaussian moments). *if $x \sim \mathcal{N}(0, 1)$ is a Gaussian variable, then for $p \in N$, $\mathbb{E}_x[x^{2p}] = \frac{(2p)!}{2^p(p!)} \leq 2^p p!$; $\mathbb{E}_x[x^{2p+1}] = 0$.*

**Definition 6** (Hermite polynomials). *In this paper, we use the normalized Hermite polynomials, which are univariate polynomials which form an orthogonal polynomial basis under the normal distribution. Specifically, they are defined by the following equality*

$$H_n(x) = \frac{(-1)^n e^{\frac{x^2}{2}}}{\sqrt{n!}} \left( \frac{d^n e^{-\frac{x^2}{2}}}{dx^n} \right)$$

The Hermite polynomials in the above definition forms a set of orthonormal basis of polynomials in the standard Normal distribution. For a polynomial $f : \mathbb{R}^n \to \mathbb{R}$, let $f(x) = \sum_{i \in I_n^{\leq p}} f_i^M \prod_{j=1}^{n} x_j^{i_j}$ and

$f(x) = \sum_{i \in I_n^{\leq p}} f_i^H \prod_{j=1}^{n} H_{i_j}(x_j)$ be its expansions in the basis of monomials and Hermite polynomials respectively ($H_k$ is the Hermite polynomial of order $k$). Let the index set $I_n^{\leq p} = \bigcup_{j=0}^{p} I_n^j$. We have the following propositions. The propositions are well-known and easy to prove. We include the proofs here for completeness.

**Proposition 4.** *for $i \in I_n^p$, $f_i^M = \left( \prod_{j=1}^{n} \frac{1}{\sqrt{i_j!}} \right) f_i^H$*

*Proof.* Consider $i = (i_1, i_2 \cdots i_n) \in I_n^p$, in the monomial expansion, the coefficient for the monomial $M_i = \prod_{j=1}^{n} x_j^{i_j}$ is $f_i^M$. In the Hermite expansion, since $H_n(x)$ is an order-$n$ polynomial, if the term $\prod_{j=1}^{n} H_{i'_j}(x_j)$ contain the monomial $M_i$, there must be $i'_j \geq i_j$, and therefore for $i \in I_n^p$ the only term in the Hermite expansion that contains $M_i$ is $f_i^H \prod_{j=1}^{n} H_{i_j}(x_j)$ (with $M_i$ as its highest order monomial). The coefficient for $x_j^{i_j}$ in $H_{i_j}(x_j)$ is $\frac{1}{\sqrt{i_j!}}$, and therefore $f_i^M = \left( \prod_{j=1}^{n} \frac{1}{\sqrt{i_j!}} \right) f_i^H$  $\square$

**Proposition 5.** *For $x \sim \mathcal{N}(0,1)^n$, $E_x[f] = f_{0^n}^H$, $E_x[f^2] = \sum_{i \in I_n^{\leq p}} (f_i^H)^2$ ($0^n$ refers to the index $(0, 0, 0 \cdots 0) \in I_n^0$).*

*Proof.* Firstly, let $w(x) = \frac{1}{\sqrt{2\pi}} e^{-x^2/2}$ be the PDF of $\mathcal{N}(0,1)$, then

$$\int_{-\infty}^{\infty} H_n(x) w(x) dx = \frac{(-1)^n}{\sqrt{2\pi n!}} \int_{-\infty}^{\infty} \left[ \frac{d^n e^{-x^2/2}}{dx^n} \right] dx$$

$$= \begin{cases} 0 & n \geq 1 \\ 1 & n = 0 \end{cases},$$

as a result of $\frac{d^n e^{-x^2/2}}{dx^n} \to 0$ when $x \to \pm\infty$ for $n \geq 0$. Besides,

$$\int_{-\infty}^{\infty} H_n(x) H_m(x) w(x) dx = \delta_{nm}$$

for its well-known orthogonality in Guassian distribution (with $\delta_{nm} = \mathbb{I}[n = m]$ as the Kronecker function). Therefore,

$$E_x[f] = \sum_{i \in I_n^{\leq p}} f_i^H \prod_{j \in [n]} \int_{-\infty}^{\infty} H_{i_j}(x_j) w(x_j) dx_j$$

$$= \sum_{i \in I_n^{\leq p}} f_i^H \prod_{j \in [n]} \mathbb{I}[i_j = 0]$$

$$= f_{0^n}^H,$$

$$E_x[f^2] = \sum_{i,i' \in I_n^{\leq p}} f_i^H f_{i'}^H \prod_{j \in [n]} \int_{-\infty}^{\infty} H_{i_j}(x_j) H_{i'_j}(x_j) w(x_j) dx_j$$

$$\begin{aligned} &= \sum_{i,i' \in I_n^{\leq p}} f_i^H f_{i'}^H \prod_{j \in [n]} \mathbb{I}[i_j = i_j'] \\ &= \sum_{i \in I_n^{\leq p}} (f_i^H)^2. \end{aligned}$$

$\square$

Then, we have the following lemma that lower bounds the variance of a polynomial with some structure. Given the following lemma, we can apply the anti-concentration results in the proof of Theorem 5 and 6.

**Lemma 9** (Variance). *Let $f(x) = f(x_1, x_2 \cdots x_d)$ be a homogeneous multivariate polynomial of degree $p$, then there is a symmetric tensor $M \in X_d^p$ that $f(x) = \langle M, x^{\otimes p} \rangle$. For all $x_0 \in \mathbb{R}^d$, when $x \sim \mathcal{N}(0,1)^d$,*

$$\mathrm{Var}_x[f(x_0 + x)] \geq \|M\|_{rv}^2$$

*Proof.* We can view $f(x_0 + x)$ as a polynomial with respect to $x$ and let $f_i^M$ and $f_i^H$ be the coefficients of its expansion in the monomial basis and Hermite polynomial basis respectively (with variable $x$). It's clear to see that $(f_i^M | i \in I_n^p)$ is the reduced vectorized form of $M$. From the Proposition 4 and 5, we have

$$\begin{aligned} \mathrm{Var}[f(x_0 + x)] &= \mathbb{E}[f(x_0 + x)^2] - \mathbb{E}[f(x_0 + x)]^2 \\ &= \sum_{i \in I_n^{\leq p} \setminus 0^n} (f_i^H)^2 \\ &\geq \sum_{i \in I_n^p} (f_i^H)^2 \geq \sum_{i \in I_n^p} (f_i^M)^2 \\ &= \|M\|_{\mathrm{rv}}^2. \end{aligned}$$

$\square$

We also need a variance bound for two sets of random variables

**Lemma 10.** *Let $f(x) = f(x_1, x_2 \cdots x_d)$ be a homogeneous multivariate polynomial of degree $2p$, then there is a symmetric tensor $M \in X_n^p$ that $f(x) = \langle M, x^{\otimes 2p} \rangle$. For all $u_0, v_0 \in \mathbb{R}^d$, when $u, v \sim \mathcal{N}(0, I_d)$, we have*

$$\mathrm{Var}_{u,v}[\langle M, (u_0 + u)^{\otimes p} \otimes (v_0 + v)^{\otimes p} \rangle] \geq \frac{1}{(2p)!} \|M\|_{rv}^2$$

*Proof.* The proof is similar to Lemma 9. We can view $\langle M, (u_0 + u)^{\otimes p} \otimes (v_0 + v)^{\otimes p} \rangle$ as a degree-$2p$ polynomial $g$ over $2d$ variables $(u, v)$. Therefore by Lemma 9 the variance would be at least the rv-norm of $g$. Note that every element (monomial in the expansion) in $M$ corresponds to at least one element in $g$, and the ratio of coefficient in the correspnding rv-basis is bounded by $(2p)!$, therefore $\|g\|_{rv} \geq \frac{1}{(2p)!} \|M\|_{rv}$, and the lemma follows from Lemma 9. $\square$

### E.2 Proof of Theorem 5

In this section, we give the formal proof of Theorem 5. First recall the setting of Theorem 5: we add a small independent Gaussian perturbation $\tilde{x} \sim \mathcal{N}(0, v)^d$ on each sample $x$, and denote $\bar{x} = x + \tilde{x}$. The output of the first layer is $\{z_j\}$ where $z_j(i) = (r_i^\top \bar{x}_j)^p$.

Our goal is to prove that $\{z_j\}$'s satisfy the conditions required by Theorem 4, in particular, the matrix $Z = [z_1^{\otimes 2}, ..., z_n^{\otimes 2}]$ has full column rank and a bound on smallest singular value. To do that, note that if we let $\bar{X} = [\bar{x}_1^{\otimes 2p}, \bar{x}_2^{\otimes 2p} \cdots \bar{x}_n^{\otimes 2p}]$ be the order-$2p$ perturbed data matrix, and $Q$ be a matrix whose $i$-th row is equal to $r_i^{\otimes p}$, then we can write $Z = (Q \otimes Q)\bar{X}$.

We first show an auxiliary lemma which helps us to bound the smallest singular value of the output matrix $(Q \otimes Q)\bar{X}$, and then we present our proof for Lemma 2.

Generally speaking, the proof of Lemma 2 consists of the lower bound of the *Leave-one-out distance* by the anti-concentration property of polynomials and the use of Lemma 8 to bridge the *Leave-one-out distance* and the smallest singular value.

**Lemma 11.** *Let $M$ be a $k$-dimensional subspace of the symmetric subspace of $X_d^p$, and let $Proj_M$ be the projection into $M$. For any $x \in R^d$ with pertubation $\tilde{x} \sim \mathcal{N}(0, v)^d$, $\bar{x} = x + \tilde{x}$,*

$$\Pr\left\{\|Proj_M \bar{x}^{\otimes p}\|_2 < \left(\frac{k}{(2p)!}\right)^{1/4} v^{\frac{p}{2}}\varepsilon\right\} < O(p)\varepsilon^{1/p}.$$

*Proof.* Let $m_1, m_2 \cdots m_k \in X_d^p$ be a set of orthonormal (in $T_d^p$ as a Euclidean space) basis that spans $M$, and each $m_i$ is symmetric. Then $\|\text{Proj}_M \bar{x}^{\otimes p}\|_2 = \sqrt{\sum_{i=1}^{k}\langle m_i, \bar{x}^{\otimes p}\rangle^2}$. Let $g(x) = \sum_{i=1}^{k}\langle m_i, x^{\otimes p}\rangle^2 = \langle \sum_{i=1}^{k} m_i^{\otimes 2}, x^{\otimes 2p}\rangle$, then $g(x)$ is a homogeneous polynomial with order $2p$. For any initial value $x$, if $\bar{x} = x + \tilde{x}$, then $\frac{1}{\sqrt{v}}\bar{x} = \frac{1}{\sqrt{v}}x + \frac{1}{\sqrt{v}}\tilde{x}$ is a vector where the random part $\frac{1}{\sqrt{v}}\tilde{x} \sim \mathcal{N}(0, 1)^n$. Therefore by Lemma 9

$$\text{Var}_x\left[g\left(\frac{1}{\sqrt{v}}\bar{x}\right)\right] \geq \|\sum_{i=1}^{k} m_i^{\otimes 2}\|_{\text{rv}}^2 \geq \frac{1}{(2p)!}\|\sum_{i=1}^{k} m_i^{\otimes 2}\|_2^2 = \frac{k}{(2p)!}.$$

Hence from Proposition 2 we know that, when $\hat{x} \sim \mathcal{N}(0, vI)$,

$$\Pr\left\{\|\text{Proj}_M \bar{x}^{\otimes p}\|_2 < \left(\frac{k}{(2p)!}\right)^{1/4} v^{p/2}\varepsilon\right\} = \Pr\left\{\left|\sqrt{\frac{(2p)!}{k}}g(\frac{\bar{x}}{\sqrt{v}})\right| < \varepsilon^2\right\} \leq O(p)\varepsilon^{1/p}.$$

$\square$

**Lemma 12.** *Let $M$ be a $k$-dimensional subspace of the symmetric subspace of $X_d^{2p}$, and let $Proj_M$ be the projection into $M$. For any $x, y \in \mathbb{R}^d$ with pertubation $\tilde{x}, \tilde{y} \sim \mathcal{N}(0, v)^d$, $\bar{x} = x + \tilde{x}$, and $\bar{y} = y + \tilde{y}$, there is*

$$\Pr\left\{\|Proj_M(\bar{x}^{\otimes p} \otimes \bar{y}^{\otimes p})\|_2 < \left(\frac{k}{((4p)!)^2}\right)^{1/4} v^p\varepsilon\right\} < O(p)\varepsilon^{1/2p}.$$

*Proof.* The proof here is similar to that of Lemma 11. Let $m_1, m_2 \cdots m_k \in X_d^{2p}$ be a set of orthonormal (in $T_d^{2p}$ as a Euclidean space) basis that spans $M$, and each $m_i$ is symmetric. Then $\|\text{Proj}_M(\bar{x}^{\otimes p} \otimes \bar{y}^{\otimes p})\|_2 = \sqrt{\sum_{i=1}^{k}\langle m_i, (\bar{x}^{\otimes p} \otimes \bar{y}^{\otimes p})\rangle^2}$. Let $g(x, y) = \sum_{i=1}^{k}\langle m_i, (x^{\otimes p} \otimes y^{\otimes p})\rangle^2 = \langle \sum_{i=1}^{k} m_i^{\otimes 2}, (x^{\otimes p} \otimes y^{\otimes p} \otimes x^{\otimes p} \otimes y^{\otimes p})\rangle = \langle \sum_{i=1}^{k} m_i^{(2)}, (x^{\otimes 2p} \otimes y^{\otimes 2p})\rangle$ for some tensor $m_i^{(2)}$, then $g(x)$ is a homogeneous polynomial with order $4p$. Notice that $\|m_i^{(2)}\|_2 = \|m_i^{\otimes 2}\|_2$ by a change of coordinate. For any initial value $x$ and $y$, if $\bar{x} = x + \tilde{x}$ and $\bar{y} = y + \tilde{y}$, then $\frac{1}{\sqrt{v}}\bar{x}$ and $\frac{1}{\sqrt{v}}\bar{y}$ are vectors where the random part $\frac{1}{\sqrt{v}}\tilde{x}, \frac{1}{\sqrt{v}}\tilde{y} \sim \mathcal{N}(0, 1)^n$. Therefore by Lemma 10,

$$\text{Var}_{x,y}\left[g\left(\frac{1}{\sqrt{v}}\bar{x}, \frac{1}{\sqrt{v}}\bar{y}\right)\right] \geq \frac{1}{(4p)!}\|\sum_{i=1}^{k} m_i^{(2)}\|_{\text{rv}}^2$$

$$\geq \frac{1}{((4p)!)^2}\|\sum_{i=1}^{k} m_i^{(2)}\|_2^2$$

$$= \frac{1}{((4p)!)^2}\|\sum_{i=1}^{k} m_i^{\otimes 2}\|_2^2$$

$$= \frac{k}{((4p)!)^2}.$$

Hence from Proposition 2 we know that, when $\hat{x} \sim \mathcal{N}(0, vI)$,

$$\Pr\left\{ \|\mathrm{Proj}_M(\bar{x}^{\otimes p} \otimes \bar{y}^{\otimes p})\|_2 < \left( \frac{k}{((4p)!)^2} \right)^{1/4} v^p \varepsilon \right\}$$

$$= \Pr\left\{ \left| \frac{(4p)!}{\sqrt{k}} g\left( \frac{\bar{x}}{\sqrt{v}}, \frac{\bar{y}}{\sqrt{v}} \right) \right| < \varepsilon^2 \right\} \leq O(p)\varepsilon^{1/2p}.$$

$\square$

Then we can show Lemma 2 as follows.

**Lemma 2.** *Suppose $k \leq O_p(d^p)$ and $\binom{k+1}{2} > n$, let $\bar{x}_j = x_j + \tilde{x}_j$ be the perturbed input in the smoothed analysis setting, where $\tilde{x}_j \sim \mathcal{N}(\mathbf{0}, v\mathbf{I})$, let $\{z_1, z_2, ..., z_n\}$ be the output of the first layer on the perturbed input $((z_j)_i = (r_i^T \bar{x}_j)^p)$. Let $Z \in \mathbb{R}^{k^2 \times n}$ be the matrix whose $j$-th column is equal to $z_j^{\otimes 2}$, then with probability at least $1 - \delta$, the smallest singular value of $Z$ is at least $\Omega_p(v^p \delta^{4p}/n^{2p+1/2} k^{4p})$.*

Actually, we show a more formal version which also states the dependency on $p$.

**Lemma 13** (Smallest singular value for $(Q \otimes Q)\bar{X}$ with pertubation)**.** *With $Q$ being the $k \times d^p$ matrix defined as $Q = [r_1^{\otimes p}, r_2^{\otimes p} \cdots r_k^{\otimes p}]^T$ ($r_i \sim \mathcal{N}(\mathbf{0}, I)$), with pertubed $\bar{X} = [\bar{x}_1^{\otimes 2p}, \bar{x}_2^{\otimes 2p} \cdots \bar{x}_n^{\otimes 2p}]$ ($\bar{x}_i = x_i + \tilde{x}_i$), and with $Z = (Q \times Q)\bar{X}$, when $\tilde{x}_i$ is drawn from i.i.d. Gaussian Distribution $\mathcal{N}(\mathbf{0}, vI)$, for $2\sqrt{n} \leq k \leq \frac{D_d^{2p}}{D_d^p \binom{2p}{p}} = O_p(d^p)$, with overall probability $\geq 1 - O(p\delta)$, the smallest singular value*

$$\sigma_{\min}(Z) \geq \left( \frac{[D_d^{2p} - k D_d^p \binom{2p}{p}][\binom{k+1}{2} - n]}{[(4p)!]^3} \right)^{1/4} \frac{v^p \delta^{4p}}{n^{2p+1/2} k^{4p}} \tag{4}$$

*Proof.* First, we show that with high probability, the projection of rows of $Q \otimes Q$ in the space of degree $2p$ symmetric polynomials (in this proof we abuse the notation $\mathrm{Proj}_{X_d^{2p}}(Q \otimes Q)$ to denote the matrix with rows being the projection of rows of $Q \otimes Q$ onto the space in question) has rank $k_2 := \binom{k+1}{2}$, and moreover give a bound on $\sigma_{k_2}(\mathrm{Proj}_{X_d^{2p}}(Q \otimes Q))$.

We do this by bounding the leave one out distance of the rows of $\mathrm{Proj}_{X_d^{2p}}(Q \otimes Q)$, note that we only consider rows $(i, j)$ as $\mathrm{Proj}_{X_d^{2p}}(r_i^{\otimes p} \otimes r_j^{\otimes p})$ where $1 \leq i \leq j \leq k$ (this is because the $(i, j)$ and $(j, i)$-th row of $\mathrm{Proj}_{X_d^{2p}}(Q \otimes Q)$ are clearly equal).

The main difficulty here is that different rows of $\mathrm{Proj}_{X_d^{2p}}(Q \otimes Q)$ can be correlated. We solve this problem using a technique similar to Ma et al. (2016).

For any $1 \leq i \leq j \leq k$, fix the randomness for $r_l$ where $l \neq i, j$. Consider the subspace $S_{(i,j)} := \mathrm{span}\{\mathrm{Proj}_{X_d^{2p}}(r_l^{\otimes p} \otimes x^{\otimes p}), x \in \mathbb{R}^d, l \neq i, j\}$. The dimension of this subspace is bounded by $k \cdot D_d^p \cdot \binom{2p}{p}$ (as there are $\binom{2p}{p}$ ways to place $p$ copies of $r_l$ and $p$ copies of $x$). Note that any other row of $\mathrm{Proj}_{X_d^{2p}}(Q \otimes Q)$ must be in this subspace.

Now by Lemma 12, we know that the projection of row $(i, j)$ onto the orthogonal subspace of $S_{(i,j)}$ has norm $\left( \frac{D_d^{2p} - k D_d^p \binom{2p}{p}}{((4p)!)^2} \right)^{1/4} \varepsilon$ with probability $O(p)\epsilon^{1/2p}$. Thus by union bound on all the rows, with probability at least $1 - O(p\delta)$, the leave-one-out distance is at least

$$l(\mathrm{Proj}_{X_d^{2p}}(Q \otimes Q)) \geq \left( \frac{D_d^{2p} - k D_d^p \binom{2p}{p}}{((4p)!)^2} \right)^{1/4} \left( \frac{\delta}{\binom{k+1}{2}} \right)^{2p},$$

and by Lemma 8 the minimal absolute singular value $\sigma_{\min}(\text{Proj}_{X_d^{2p}}(Q \otimes Q)) \geq \dfrac{l\left(\text{Proj}_{X_d^{2p}}(Q \otimes Q)\right)}{\sqrt{\binom{k+1}{2}}}$.

Next, let $V(Q \otimes Q)$ be the rowspace of $\text{Proj}_{X_d^{2p}}(Q \otimes Q)$, which as we just showed has dimension $\binom{k+1}{2}$. We wish to show that the projections of columns of $X$ in $V(Q \otimes Q)$ have a large leave-one-out distance, and thus $(Q \otimes Q)X$ has a large minimal singular value.

Actually for each $i$, the subspace (which for simplicity will be denoted as $V_{-i}(Q \otimes Q)$) of $V(Q \otimes Q)$ orthogonal to $span\{\bar{x}_j^{\otimes 2p}|j \neq i\}$ has dimension $\binom{k+1}{2} - n + 1$ almost surely, and therefore by Lemma 11 and union bound, with probability $1 - O(p)\tau^{1/2p}n = 1 - O(p\delta)$, for all $i$,

$$\|P_{V_{-i}(Q \otimes Q)}(x_i^{\otimes 2p})\|_2 = \mathbb{E}\left[\|P_{V_{-i}(Q \otimes Q)}(x_i^{\otimes 2p})\|_2 \Big| \{\bar{x}_j | j \neq i\}\right] \geq \left(\frac{\binom{k+1}{2} - n}{(4p)!}\right)^{1/4} v^p \tau,$$

thus with probability $1 - O(p\delta)$, for any vector $c \in R^n$ with $\|c\|_2 = 1$, let $i^* = argmax_i|c_i|$, $|c_{i^*}| \geq \frac{1}{\sqrt{n}}$, and

$$\begin{aligned}
\|(Q \otimes Q)\hat{X}c\|_2 \quad &\geq \sigma_{\min}(\text{Proj}_{X_d^{2p}}Q \otimes Q)|c_{i^*}|\|\text{Proj}_{V(Q \otimes Q)}\hat{X}\tfrac{c}{|c_{i^*}|}\|_2 \\
&\geq \frac{\sigma_{\min}(\text{Proj}_{X_d^{2p}}Q \otimes Q)}{\sqrt{n}}\|\text{Proj}_{V_{-i^*}(Q \otimes Q)}(x_{i^*}^{\otimes 2p})\|_2 \\
&\geq \left(\frac{[D_d^{2p} - kD_d^p\binom{2p}{p}][\binom{k+1}{2} - n]}{[(4p)!]^3}\right)^{1/4} \frac{v^p \delta^{4p}}{n^{2p+1/2}k^{4p}}
\end{aligned}$$

And therefore we will get Lemma 13. $\qquad\square$

A minor requirement of on $z_j$'s is that they all have bounded norm. This is much easier to prove:

**Lemma 14** (Norm upper bound for $Q\bar{x}^{\otimes p}$)**.** *Suppose that $\|x_j\|_2 \leq B$ for all $j \in [n]$ and $\bar{x}_j = x_j + \tilde{x}_j$ where $\tilde{x}_j \sim \mathcal{N}(\mathbf{0}, vI)$. Same as the previous notation, $Q = [r_1^{\otimes p}, \dots, r_k^{\otimes p}]^T \in \mathbb{R}^{k \times d^p}$. Then with probability at least $1 - \dfrac{\delta}{\sqrt{2\pi \ln((k+n)d\delta^{-1/2})}(k+n)d}$, for all $i \in [n]$, we have*

$$\|Q\bar{x}_i^{\otimes p}\|_2 \leq \sqrt{k}\left(2(B + 2\sqrt{vd\ln((k+n)d\delta^{-1/2})})\sqrt{d\ln((k+n)d\delta^{-1/2})}\right)^p.$$

*Proof.* First we have, for a standard normal random variable $N \sim \mathcal{N}(0, 1)$, we have

$$\Pr\{|N| \geq x\} \leq \frac{\sqrt{2}}{\sqrt{\pi}x}e^{-\frac{x^2}{2}}.$$

Then, apply the union bound, we have with probability at least $1 - \dfrac{\delta}{\sqrt{2\pi \ln((k+n)d\delta^{-1/2})}(k+n)d}$, for all $l \in [k], i \in d, j \in [n], \ell \in d, \delta < 1$, we have

$$|(r_l)_i| \leq 2\sqrt{\ln((k+n)d\delta^{-1/2})}, |(\tilde{x}_j)_\ell| \leq 2\sqrt{v\ln((k+n)d\delta^{-1/2})}.$$

Then for all $j \in [n]$, we have

$$\|\bar{x}\|_2 \leq \|x\|_2 + \|\tilde{x}\|_2 \leq B + 2\sqrt{vd\ln((k+n)d\delta^{-1/2})}.$$

If for all $i \in [d], l \in [k], |(r_j)_i| < 2\sqrt{\ln((k+n)d\delta^{-1/2})}$, then for any $\bar{x}$ such that $\|\bar{x}\| \leq B + 2\sqrt{vd\ln((k+n)d\delta^{-1/2})}$ and any $l \in [k]$, we have

$$\begin{aligned}
\left|\left((r_l)^{\otimes p}\right)^T \bar{x}^{\otimes p}\right| &= |(r_l^T\bar{x})^p| \\
&\leq (\|r_l\| \cdot \|\bar{x}\|)^p \\
&\leq \left(2(B + 2\sqrt{vd\ln((k+n)d\delta^{-1/2})})\sqrt{d\ln((k+n)d\delta^{-1/2})}\right)^p.
\end{aligned}$$

Then we have

$$||Q\bar{x}^{\otimes p}||_2 \leq \sqrt{k}\left(2(B + 2\sqrt{vd\ln((k+n)d\delta^{-1/2})})\sqrt{d\ln((k+n)d\delta^{-1/2})}\right)^p.$$

□

Then combined with the previous lemmas which lower bound the smallest singular value(Lemma 13) and upper bound the norm(Lemma 14) of the outputs of the random feature layer and Theorem 4, we have the following Theorem 5.

**Theorem 5** (Main theorem for 3-layer NN). *Suppose the original inputs satisfy $\|x_j\|_2 \leq 1$, $|y_j| \leq 1$, inputs $\bar{x}_j = x_j + \tilde{x}_j$ are perturbed by $\tilde{x}_j \sim \mathcal{N}(0, vI)$, with probability $1 - \delta$ over the random initialization, for $k = 2\lceil\sqrt{n}\rceil$, perturbed gradient descent on the second layer weights achieves a loss $f(W^*) \leq \epsilon$ in $O_p(1) \cdot \frac{(n/v)^{O(p)}}{\epsilon^{5/2}}\log^4(n/\epsilon)$ iterations.*

*Proof.* From the above lemmas, we know that with respective probability $1 - o(1)\delta$, after the random featuring, the following happens:

1. $\sigma_{\min}((Q \otimes Q)\bar{X}) \geq \left(\frac{[D_d^{2p} - kD_d^p\binom{2p}{p}][\binom{k+1}{2}] - n]}{[(4p)!]^3}\right)^{1/4} \frac{v^p\delta^{4p}}{p^{4p}n^{2p+1/2}k^{4p}}$

2. $\|Q\bar{x}_j^{\otimes p}\|_2 \leq \sqrt{k}\left(2(B + 2\sqrt{vd\ln((k+n)d\delta^{-1/2})})\sqrt{d\ln((k+n)d\delta^{-1/2})}\right)^p$ for all $j \in [n]$.

Thereby considering the PGD algorithm on $W$, since the random featuring outputs $[(r_i^T\bar{x}_j)^p] = Q[\bar{x}_j^{\otimes p}]$ has $[(r_i^T\bar{x}_j)^{2p}] = (Q \otimes Q)\bar{X}$, from Theorem 4, given the singular value condition and norm condition above we obtain the result in the theorem. □

### E.3 PROOF OF THEOREM 6

In this section, we show the proof of Theorem 6. In the setting of Theorem 6, we do not add perturbation onto the samples, and the only randomness is the randomness of parameters in the random feature layer.

Recall that $Q \in \mathbb{R}^{k \times d^p}$ is defined as $Q = [r_1^{\otimes p}, r_2^{\otimes p} \cdots r_k^{\otimes p}]^T$. We show that: when $r_i$ is sampled from i.i.d. Normal distribution $\mathcal{N}(0, 1)^d$ and $k$ is large enough, with high probability $Q$ is robustly full column rank. Let $N_\varepsilon$ and $N_\sigma$ be respectively an $\varepsilon$-net and a $\sigma$-net of $\bar{X}_d^p$ with size $Z_\varepsilon$ and $Z_\sigma$.

The following lemmas(Lemma 15, 16 and 17) apply the standard $\varepsilon$-net argument and lead to the smallest singular value of matrix $Q$(Lemma 18). Then we will derive the smallest singular value for the matrix $(Q \otimes Q)X$(Lemma 19).

Note that unlike the $Q$ matrix in the previous section, in this section the $Q$ matrix is going to have more rows than columns, so it has full column rank (restricted to the symmetry of $Q$). The $Q$ matrix in the previous section has full row rank. This is why we could not use the same approach to bound the smallest singular value for $Q$.

**Lemma 15.** *For some constant $C$, with probability at least $1 - Z_\varepsilon\left(Cp\eta^{1/p}\right)^k$, for all $c \in N_\varepsilon$, we have*

$$\|Qc\|_2^2 \geq \frac{\eta^2}{p!}.$$

*Proof.* For any $c \in \bar{X}_d^p$, by Lemma 6, $\|c\|_{rv} \geq \frac{1}{\sqrt{p!}}$. Let $f(r) = c^T r^{\otimes p}$, then $f$ is a polynomial of degree $p$ with respect to $r$, and therefore by Lemma 9,

$$\underset{r \sim \mathcal{N}(0,1)^d}{\text{Var}}[f(r)] \geq \|c\|_{rv}^2 \geq \frac{1}{p!}.$$

Thus by Proposition 2,

$$\Pr_{r \sim \mathcal{N}(0,1)^d} \left\{ |f(r)| < \frac{\eta}{\sqrt{p!}} \right\} \leq O(p)\eta^{1/p}.$$

Therefore, as $\|Qc\|_2^2 = \sum\limits_{i=1}^{K} f(r_i)^2$,

$$\Pr_{r_1, r_2 \cdots r_K \sim \mathcal{N}(0,1)^d} \left\{ \|Qc\|_2^2 < \frac{\eta^2}{p!} \right\} \leq \Pr_{r_1, r_2 \cdots r_K \sim \mathcal{N}(0,1)^d} \left\{ \forall r_i : |f(r_i)| < \frac{\eta}{\sqrt{p!}} \right\}$$

$$\leq \left( O(p)\eta^{1/p} \right)^k.$$

Therefore for some constant $C$, for each $c \in \bar{X}_d^p$, with probability at most $\left( Cp\eta^{1/p} \right)^k$ there is $\|Qc\|_2^2 < \frac{\eta^2}{p!}$. Thus by union bound this happens for all $c \in N_\varepsilon$ with probability at most $\leq Z_\varepsilon \left( Cp\eta^{1/p} \right)^k$, and thereby the proof is completed. $\qquad\square$

**Lemma 16.** *For $\tau > 0$, with probability $1 - O\left( (Z_\sigma \left( \frac{\sqrt{k}}{\tau} \right)^{1/p} k e^{-\frac{1}{2} \left( \frac{\tau}{\sqrt{k}} \right)^{2/p}} \right)$, for each $c \in N_\sigma$, $\|Qc\|_2 \leq \tau$.*

*Proof.* For any $c \in \bar{X}_d^p$,

$$\Pr_Q \left\{ \|Qc\|_2^2 > \tau^2 \right\} \leq \Pr_{r_1, r_2 \cdots r_k \sim \mathcal{N}(0,1)^d} \left\{ \exists i : |c^T r_i^{\otimes p}| > \frac{\tau}{\sqrt{k}} \right\}$$

$$\leq k \Pr_{r \sim \mathcal{N}(0,1)^d} \left\{ |c^T r^{\otimes p}| > \frac{\tau}{\sqrt{k}} \right\}.$$

Furthermore,

$$\Pr_{r \sim \mathcal{N}(0,1)^d} \left\{ |c^T r^{\otimes p}| > \frac{\tau}{\sqrt{k}} \right\} \leq \Pr_{r \sim \mathcal{N}(0,1)^d} \left\{ \|c\|_2 \|r\|_2^p > \frac{\tau}{\sqrt{k}} \right\}$$

$$= \Pr_{r \sim \mathcal{N}(0,1)^d} \left\{ \|r\|_2 > \left( \frac{\tau}{\sqrt{k}} \right)^{1/p} \right\}$$

$$\leq O\left( \left( \frac{\sqrt{k}}{\tau} \right)^{1/p} e^{-\frac{1}{2} \left( \frac{\tau}{\sqrt{k}} \right)^{2/p}} \right)$$

Therefore for the $\sigma$-net $N_\sigma$, with a union bound we know with probability at least $1 - O\left( (Z_\sigma \left( \frac{\sqrt{k}}{\tau} \right)^{1/p} k e^{-\frac{1}{2} \left( \frac{\tau}{\sqrt{k}} \right)^{2/p}} \right)$, for all $c \in N_\sigma$, $\|Qc\|_2^2 \leq \tau^2$. $\qquad\square$

**Lemma 17.** *For $\sigma < 1$, $\tau > 0$, with probability at least $1 - O\left( Z_\sigma \left( \frac{\sqrt{k}}{\tau} \right)^{1/p} k e^{-\frac{1}{2} \left( \frac{\tau}{\sqrt{k}} \right)^{2/p}} \right)$, we have for each $c \in \bar{X}_d^p$, $\|Qc\|_2 \leq \frac{\tau}{1-\sigma}$.*

*Proof.* We first show that give $N_\sigma$, for each $c \in \bar{X}_d^p$, we can find $c_1, c_2, c_3 \cdots \in N_\sigma$ and $a_1, a_2, a_3 \cdots \in \mathbb{R}$ such that

$$c = \sum_{i \geq 1} a_i c_i,$$

and that $a_1 = 1$, $0 \leq a_i \leq \sigma a_{i-1}$ ($i \geq 2$). Thus $a_i \leq \sigma^{i-1}$.

In fact, we can construct the sequence by induction. Let $I : \bar{X}_d^p \to N_\sigma$ that

$$I(x) = \operatorname*{argmin}_{y \in N_\sigma} \|y - x\|_2.$$

We take $c_1 = I(c)$, $a_1 = 1$, and recursively

$$a_i = \left\| c - \sum_{j=1}^{i-1} a_j c_j \right\|_2, \quad c_i = I \left( \frac{c - \sum\limits_{j=1}^{i-1} a_j c_j}{a_i} \right).$$

By definition, for any $c \in \bar{X}_d^p$, $\|c - I(c)\|_2 \leq \sigma$, and therefore

$$\left\| \frac{c - \sum_{j=1}^{i-1} a_j c_j}{a_i} - c_i \right\|_2 \leq \sigma,$$

which shows that $0 \leq a_{i+1} = \|c - \sum\limits_{j=1}^{i} a_j c_j\|_2 \leq \sigma a_i$, and by induction $a_i \leq \sigma^{i-1}$.

We know from Lemma 16 that with probability at least $1 - O\left( Z_\sigma \left( \frac{\sqrt{k}}{\tau} \right)^{1/p} k e^{-\frac{1}{2} \left( \frac{\tau}{\sqrt{k}} \right)^{2/p}} \right)$, for all $c_i \in N_\sigma$, $\|Q c_i\|_2 \leq \tau$, and therefore

$$\|Qc\|_2 \leq \sum_{i \geq 1} a_i \|Q c_i\|_2 \leq \sum_{i \geq 1} \sigma^{i-1} \tau = \frac{\tau}{1 - \sigma}.$$

$\square$

**Lemma 18** (least singular value of $Q$)**.** *If $Q$ is the $k \times d^p$ matrix defined as $Q = [r_1^{\otimes p}, r_2^{\otimes p} \cdots r_k^{\otimes p}]^T$ with $r_i$ drawn i.i.d. from Gaussian Distribution $\mathcal{N}(0, I)$, then there exists constant $G_0 > 0$ that for $k = \alpha p D_d^p$ ($\alpha > 1$), with probability at least $1 - o(1)\delta$, the rows of $Q$ will span $X_d^p$, and for all $c \in \bar{X}_d^p$,*

$$\|Qc\|_2 \geq \Omega \left( \frac{\delta^{\left( \frac{1}{(\alpha-1)D_d^p} \right)}}{\left( p^p \sqrt{p!} \right)^{\frac{\alpha}{\alpha-1}} \left( k(G_0 p \ln p D_d^p)^p \right)^{\frac{1}{2(\alpha-1)}}} \right) = \Omega_p \left( \frac{\delta^{\left( \frac{1}{(\alpha-1)D_d^p} \right)}}{k^{\frac{p+1}{2(\alpha-1)}}} \right),$$

*where $\Omega_p$ is the big-$\Omega$ notation that treats $p$ as a constant.*

*Proof.* We show that with high probability, for all $c \in \bar{X}_d^p$, $\|Qc\|_2^2 = \sum\limits_{i=1}^{k} \left( [r_i^{\otimes p}]^T c \right)^2$ is large. To do this we will adopt an $\varepsilon$-net argument over all possible $c$.

First, we take the parameters

$$\sigma = \frac{1}{10}, \quad \tau = \sqrt{k \left( 2 \log \frac{Z_\sigma k}{\delta} \right)^p}, \quad \text{and} \quad \varepsilon = c_0 \frac{\delta^{\left( \frac{1}{(\alpha-1)D_d^p} \right)}}{\left( \tau p^p \sqrt{p!} \right)^{\frac{\alpha}{\alpha-1}}},$$

for small constant $c_0$ such that $c_0 C^p D^{\frac{1}{(\alpha-1)D}} \ll 1$, and $\eta = \frac{20}{9} \varepsilon \tau \sqrt{p!}$. From Lemma 15 and 17, we know that with probability at least

$$1 - Z_\varepsilon \left( c p \eta^{1/p} \right)^k - O\left( Z_\sigma \left( \frac{\sqrt{k}}{\tau} \right)^{1/p} k e^{-\frac{1}{2} \left( \frac{\tau}{\sqrt{k}} \right)^{2/p}} \right)$$

$$= 1 - O\left( c_0^{(\alpha-1)D_d^p} 2^{D_d^p} C^k D \delta \right) - O\left( \frac{\delta}{\sqrt{2 \log \frac{Z_\sigma k}{\delta}}} \right)$$

$$= 1 - o(1)\delta,$$

the following holds true:

1. $\forall c_i \in N_\varepsilon$, $\|Q c_i\|_2 \geq \frac{\eta}{\sqrt{p!}}$;

2. $\forall c \in \bar{X}_d^p, \|Qc\|_2 \leq \frac{\tau}{1-\sigma} = \frac{\eta}{2\varepsilon\sqrt{p!}}$.

Therefore for any $c \in \bar{x}_d^p$, let $i^* = \underset{i:c_i \in N_\varepsilon}{\operatorname{argmin}} \|c - c_i\|_2$, we know

$$
\begin{aligned}
\|Qc\|_2 &\geq \|Qc_i\|_2 - \|Q(c - c_i)\|_2 \\
&\geq \frac{\eta}{\sqrt{p!}} - \|c - c_i\|_2 \left\| Q\frac{c - c_i}{\|c - c_i\|_2} \right\|_2 \\
&\geq \frac{\eta}{\sqrt{p!}} - \varepsilon\frac{\eta}{2\varepsilon\sqrt{p!}} \\
&= \frac{\eta}{2\sqrt{p!}},
\end{aligned}
$$

and by definition we know that $\lambda_{\min}(Q) \geq \frac{\eta}{2\sqrt{p!}}$. By lemma 7, with $\log Z_\sigma = O(p \ln p D_d^p) \leq G_0 p \ln p D_d^p$ for some constant $G_0$, this gives us the lemma. $\square$

**Lemma 19** (Smallest singular value for $(Q \otimes Q)X$ without pertubation). *With $Q$ being the $k \times d^p$ matrix defined as $Q = [r_1^{\otimes p}, r_2^{\otimes p} \cdots r_k^{\otimes p}]^T$, $X$ being the $d^{2p} \times n$ matrix defined as $X = [x_1^{\otimes 2p}, \ldots, x_n^{\otimes 2p}] \in \mathbb{R}^{d^{2p} \times n}$, and $Z = (Q \otimes Q)X$, for $k = \alpha p D_d^p$ ($\alpha > 1$), when $r_i$ are randomly drawn from i.i.d. Guassian distribution $\mathcal{N}(\mathbf{0}, I)$, there exists constant $G_0 > 0$ such that with probability $\geq 1 - o(1)\delta$, the smallest singular value of $Z$ satisfies*

$$
\sigma_{\min}(Z) \geq \Omega\left( \frac{\delta^{\left(\frac{2}{(\alpha-1)D_d^p}\right)}\sigma_{\min}(X)}{\left(p^p\sqrt{p!}\right)^{\frac{2\alpha}{\alpha-1}} [k(G_0 p \ln p D_d^p)^p]^{\frac{1}{(\alpha-1)}}} \right) = \Omega_p\left( \frac{\delta^{\left(\frac{2}{(\alpha-1)D_d^p}\right)}}{k^{\frac{p+1}{(\alpha-1)}}} \right)\sigma_{\min}(X) \quad (5)
$$

*(where $\Omega_p$ is the big-$\Omega$ notation that treats $p$ as a constant). Furthermore, for $k = \Omega(p^2 D_d^p)$, with high probability $1 - \delta$, $\sigma_{\min}(Z) \geq \Omega(\frac{\sigma_{\min}(X)}{k})$ (if $\delta$ is not exponentially small).*

*Proof.* From Lemma 18, with probability $\geq 1 - o(1)\delta$, for all $c \in \bar{X}_d^p$, $\|Qc\|_2 \geq \Delta = \Omega\left( \frac{\delta^{\left(\frac{1}{(\alpha-1)D_d^p}\right)}}{\left(p^p\sqrt{p!}\right)^{\frac{\alpha}{\alpha-1}} \left(k(G_0 p \ln p D_d^p)^p\right)^{\frac{1}{2(\alpha-1)}}} \right)$. Then, from linear algebra, we know for all $s \in \bar{X}_d^p \otimes \bar{X}_d^p$, $\|(Q \otimes Q)s\|_2 \geq \Delta^2$. As $\bar{X}_d^{2p} \subset \bar{X}_d^p \otimes \bar{X}_d^p$,

$$
\begin{aligned}
\sigma_{\min}(Q \otimes Q)X &= \inf_{u \in R^n, \|u\|_2=1} \|(Q \otimes Q)Xu\|_2 \\
&= \inf_{u \in R^n, \|u\|_2=1} \|(Q \otimes Q)\frac{Xu}{\|Xu\|_2}\|_2 \|Xu\|_2 \geq \Delta^2 \inf_{u \in R^n, \|u\|_2=1} \|Xu\|_2 = \Delta^2 \sigma_{\min}(X),
\end{aligned}
$$

which gives us this lemma 19. $\square$

Besides the lower bound for the smallest singular value, we also need the following lemma to show that with high probability, the norm is upper bounded.

**Lemma 20** (Norm upper bound for $Qx^{\otimes p}$). *Suppose that $\|x_i\|_2 \leq B$ for all $i \in [n]$, and $Q = [r_1^{\otimes p}, \ldots, r_k^{\otimes p}]^T \in \mathbb{R}^{k \times d^p}$. Then with probability at least $1 - \frac{\delta}{\sqrt{2\pi \ln(kd\delta^{-1/2})kd}}$, for all $i \in [n]$, we have*

$$
\|Qx_i^{\otimes p}\|_2 \leq \sqrt{k}\left(2B\sqrt{d\ln(kd\delta^{-1/2})}\right)^p.
$$

*Proof.* First we have, for a standard normal random variable $N \sim \mathcal{N}(0, 1)$, we have

$$
\Pr\{|N| \geq x\} \leq \frac{\sqrt{2}}{\sqrt{\pi}x}e^{-\frac{x^2}{2}}.
$$

Then, apply the union bound, we have

$$
\Pr\left\{\exists i \in [d], j \in [k], |(r_j)_i| \geq 2\sqrt{\ln(kd\delta^{-1/2})}\right\} \leq kd\frac{\sqrt{2}}{\sqrt{\pi}2\sqrt{\ln(kd\delta^{-1/2})}}\exp\left(-2\ln(kd\delta^{-1/2})\right)
$$

$$= \frac{\delta}{\sqrt{2\pi \ln(kd\delta^{-1/2})kd}}.$$

If for all $i \in [d], j \in [k], |(r_j)_i| < 2\sqrt{\ln(kd)}$, then for any $x$ such that $||x|| \leq B$ and any $k_0 \in [k]$, we have

$$
\begin{aligned}
\left| \left( (r_{k_0})^{\otimes p} \right)^T x^{\otimes p} \right| &= |(r_{k_0}^T x)^p| \\
&\leq (||r_{k_0}|| \cdot ||x||)^p \\
&\leq (2B\sqrt{d\ln(kd\delta^{-1/2})})^p.
\end{aligned}
$$

Then we have

$$||Qx^{\otimes p}||_2 \leq \sqrt{k}\left(2B\sqrt{d\ln(kd\delta^{-1/2})}\right)^p.$$

$\square$

Then, combining the previous lemmas and Theorem 4, we have the following Theorem 6.

**Theorem 6.** *Suppose the matrix $X = [x_1^{2p}, ..., x_n^{2p}] \in \mathbb{R}^{d^{2p} \times n}$ has full column rank, and smallest singular value at least $\sigma$. Choose $k = O_p(d^p)$, with high probability perturbed gradient descent on the second layer weights achieves a loss $f(W^*) \leq \epsilon$ in $O_p(1) \cdot \frac{(n)^{\tilde{O}(p)}}{\sigma^5 \epsilon^{5/2}} \log^4(n/\epsilon)$ iterations.*

*Proof.* From the above lemmas, we know that with respective probability $1 - o(1)\delta$, after the random featuring, the following happens:

1. There exists constant $G_0$ that $\sigma_{\min}((Q \otimes Q)X) \geq \dfrac{\delta^{\left(\frac{2}{(\alpha-1)D_d^p}\right)} \sigma_{\min}(X)}{\left(p^p \sqrt{p!}\right)^{\frac{2\alpha}{\alpha-1}} \left[k(G_0 p \ln p D_d^p)^p\right]^{\frac{1}{(\alpha-1)}}}$

2. $||Qx_j^{\otimes p}||_2 \leq \sqrt{k}(2B\sqrt{d\ln(kd\delta^{-1/2})})^p$ for all $j \in [n]$.

Thereby considering the PGD algorithm on $W$, since the random featuring outputs $[(r_i^T x_j)^p] = Q[x_j^{\otimes p}]$ has $[(r_i^T x_j)^2 p] = (Q \otimes Q)X$, from Theorem 4, given the singular value condition and norm condition above we obtain the result in the theorem. $\square$

