# OpenReview forum: "Mildly Overparametrized Neural Nets can Memorize Training Data Efficiently"
_ICLR.cc/2020/Conference — Reject_

### Official Review · AnonReviewer2 · 2019-10-22
**Official Blind Review #2**

**Rating:** 1

**Review:**

This paper studies the mildly over-parameterized neural networks. In particular, it shows that when the width is at least O(sqrt{n}) where n is the number of samples, PDG fits the training data. The analysis is done for 2-layer or 3-layer networks with (mainly) quadratic activations, with only one-layer weights (the weights before the quadratic activation) trainable. For 2-layer network, that means the first layer weights are trainable. Except for O(sqrt{n}) width, the other settings of this paper are quite restricted and it is not clear how it reflects the training of the neural networks.
    The quadratic activation is probably the main limitation of this paper, and fixing the last layer is the second major limitation. There are already a few papers studying quadratic activations, some of them cited in this paper. But the closest one is probably a missing one: Soltanolkotabi et al. https://arxiv.org/pdf/1707.04926.pdf. It also only requires O(sqrt{n}) neurons. It seems the data assumption of that paper is stronger (Gaussian input), but in terms of the major claim of this paper on mild overparameterization, that paper already made an attempt. In addition, that paper optimizes both v and W (first and second layer weights), which is more practical than this paper.
   The title on “mild over-parameterized network” is a bit misleading. The paper tries to compare with “a series of work (Du et al. (2019); Allen-Zhu et al. (2019c); Chizat & Bach (2018); Jacot et al. (2018)” which uses a large number of neurons, and thus claims “mild over-parameterization”. However, this comparison seems to be a bit  unfair, because for shallow networks there are many papers that do not require that large number of neurons, which seem to be ignored by this paper, including Brutzkus et al. arxiv:1710.10174,  Wang et al. arXiv:1808.04685, Liang et al. arXiv:1803.00909,Zhang arXiv:1806.07808 (I guess there are more). The settings of these papers are not the same as the setting of the current paper, but the differences will need to be clarified if the authors want to claim “the first step towards mild overparameterized networks”.

   The result on 3-layer projection with smoothed analysis is theoretically interesting. However, the 3-layer result is mainly for technical purpose (“restore representation power”), and does not provide much extra insight on algorithm performance --- the two-layer convergence result and the 1st layer projection are treated independently. It is not sure how useful this result will be, and how can this be extendable.

=========After rebuttal period, I found [Ref1] studied the same setting======
After the rebuttal period, I found the reference [Ref1] I mentioned earlier does not need Gaussian assumption. In fact, Theorem 2.1 applies to almost all data, thus is the SAME setting as the 2-layer part of this paper. Theorem 2.5 of [Ref1] assumed Gaussian data, but that is not the major result of [Ref1].
The 3-layer proof may still be interesting. The authors can address the differences with [Ref1] in a future version, and focus more on the 3-layer proof. Finally, it is possible that the regularizers make the proof different from [Ref1], but that requires much effort to explain (I think even there is a difference, it is probably not a major innovation).

[Ref1] Theoretical insights into the optimization landscape of
over-parameterized shallow neural networks





**Experience Assessment:**

I have published in this field for several years.

**Review Assessment: Checking Correctness Of Derivations And Theory:**

I assessed the sensibility of the derivations and theory.

**Review Assessment: Checking Correctness Of Experiments:**

I assessed the sensibility of the experiments.

**Review Assessment: Thoroughness In Paper Reading:**

I read the paper at least twice and used my best judgement in assessing the paper.

---

> ### Author Response · Authors · 2019-11-15
> **Response to Reviewer 2**
>
> Dear Reviewer 2,
>
> Thanks for your comments.
>
> [Theoretical insights into the optimization landscape of over-parameterized shallow neural networks]
> Thanks for pointing us to this result. This is clearly very relevant and we did not know about this result before, we will definitely discuss the contributions of this paper carefully in our revision. The main idea of this paper is indeed very close to our 2-layer case. However we also note some differences:
> 1. They assume that the samples are generated by a Gaussian distribution, which is a special case under our assumption. Our generalization is very crucial for us to generalize the result to 3-layers, as the output of the first layer is clearly not going to be Gaussian.
> 2. Their result characterizes the optimization landscape (which shows all local minima are globally optimal), but does not actually give an efficient algorithm for finding a local minimum. It is a common mistake to assume that the optimization landscape result directly implies existence of algorithms, however we emphasize that all existing algorithmic guarantees require the function to have smoothness and Hessian-Lipschitzness, which is *NOT* true for the objective as we noted in Section 3.2. We solve this problem by carefully adding a regularizer.
> 3. Their result also does not allow optimizing v and W (both layers' weights) simultaneously. Note that their landscape characterization requires the top-layer weights to have a certain number of positive and negative entries (which is the same as ours), but they do not prove if you train the top layer weights this condition will always be satisfied.
>
> [Other related works for 2-layers] Thanks for pointing these out. These works are in several categories and we agree it would be good to discuss some of them. Brutzkus et al. arxiv:1710.10174, Wang et al. arXiv:1808.04685 requires the data to be linearly separable, which is a very strong assumption (in their setting it's also not clear why one would need more than one neuron). Liang et al. arXiv:1803.00909 is a landscape result that does *NOT* imply efficient algorithm, there are several other landscape results (see references cited in the blog post http://www.offconvex.org/2019/06/16/modeconnectivity/) but because they don't handle high order saddle points, there is no guaranteed algorithm in these settings (we also note that the particular result of Liang et al. is also in the highly overparametrized setting by our definition, #neurons > #training sample);  Zhang arXiv:1806.07808 provides local convergence and is not overparametrizing at all. Indeed we should (and will) improve our introduction to compare our results with two sets of results - (a) the ``exact parametrization'' results (also called proper learning) where the learned network has the same size as a teacher network; and (b) the highly overparametrized results where number of neurons (as opposed to parameters) is much larger than the number of training samples. For algorithms in class (a), they often require strong distributional assumptions on the input, and even under such assumptions it is known that gradient descent does not work for more than one neuron. We have updated our intro/related work sections to clarify relationships to these works.

---

### Official Review · AnonReviewer1 · 2019-10-22
**Official Blind Review #1**

**Rating:** 3

**Review:**


This paper proves that one can design a (shallow)neural network that with a mild amount of overparametrization (e.g. the number of datapoints n is roughly less than d^2 in Theorem 1), a second-order method can reach a global minimum. In general, I think this is an interesting direction. However, I have some doubts regarding the comparison to prior work (especially regarding the results on the two-layer network), as well as some technical details that need some clarification. Regarding the quality of the writing, it’s in general ok but there are lots of grammatical mistakes, the authors should pay more attention to this. I’m not giving a high score for now but I will reconsider my review once I hear back from the authors.

Comparison to Oymak & Soltanolkotabi 2019: my understanding is that this paper prove convergence to a global minimum for a neural network where the number of parameters is only twice the number of datapoints so aren’t your results “worse” in that sense? The text in the related work seem to say the opposite, so I’m rather confused by your statement, please clarify.

Landscape: I think this discussion is largely missing in the paper but another way to prove the same result would be to focus on showing that the loss surface is “well-behaved”. In fact the paper by Soltanolkotabi et al (see reference below) proves this for a similar network with a quadratic activation function and arbitrary data. Essentially, they show that such network obeys the following two properties:
There are no spurious local minima, i.e. all local minima are global.
All saddle points have a direction of strictly negative curvature
My understanding is that they prove this for a parametrization regime where n=d^2, isn’t this the same regime as proved in your results?
Mahdi Soltanolkotabi, Adel Javanmard, and Jason D Lee. Theoretical insights into the optimization landscape of over-parameterized shallow neural networks.

Three-layer neural net
1) This network uses activation function of the form x^p. You wrote “for some constant p”, is there any specific lower bound on p? I’m also surprised that one would allow large values of p as such functions have a saddle at x=0 with a large region with low gradient magnitudes around it.
2) Limitation quadratic activation function. You say “To address this problem, the three-layer neural net in this section uses the first-layer as a random mapping of the input“. How is this helping with changing the activation function?
3) You have to fix part of the weights of the network, this seems to be a limitation of the analysis that should be more clearly highlighted and better contrasted to what has been done in prior work.
4) I think it would be interesting for the reader to focus more on the three layer network (instead of the two-layer one whose analysis is rather simple) and provide a more detailed proof sketch.

Paper organization
The most interesting result of the paper is the one about the three-layer network but the entire analysis is relegated to the appendix. I feel it would be worth trying to provide a rough proof sketch in the main paper to highlight the difficulty of the analysis.

Proof Lemma 2
Alternatively to the current proof, couldn’t you differentiate the second term (2/n \sum_j (\sum …)^2 ) w.r.t. z and show it is zero at the argmax_z? Isn’t this what your result says?

Experiments
Consider repeating the experiments a few times and showing the average.

Minor: Low loss vs perfect fitting: I couldn’t find any discussion about this but the authors seem to assume that a zero loss directly implies that the training data is fit perfectly. In the case n=d, there exists a function that yields a perfect fit of the data and therefore a larger network should be able to represent this function. Perhaps it would be worth writing this down.

More minor comments
- Figure experiments: please use a log scale
- Formula top of page 16: should be z_k on the left and right of the term inside the bracket
- missing citation top of page 24: “e.g. in”


**Experience Assessment:**

I have published one or two papers in this area.

**Review Assessment: Checking Correctness Of Derivations And Theory:**

I carefully checked the derivations and theory.

**Review Assessment: Checking Correctness Of Experiments:**

I assessed the sensibility of the experiments.

**Review Assessment: Thoroughness In Paper Reading:**

I read the paper thoroughly.

---

> ### Author Response · Authors · 2019-11-15
> **Response to Reviewer 1**
>
> Dear Reviewer 1,
>
> Thanks for your comments. We will answer your questions one by one.
>
> [Comparison with Oymak & Soltanolkotabi 2019] The result of the paper is indeed hard to parse so let us point more directly to their discussions. Their result (Theorem 2) requires kd >= \kappa^2 n^2 (even if we assume optimistically for several other terms in the formula), where n is the number of samples, d is the dimension, and k is the number of neurons. As they discussed (on top of page 6) in the best case scenario \kappa is around \sqrt{d/n}, this means one would need kd >= dn, or k >= n. Here the number of *neurons* is much larger than the number of samples, which is exactly the highly overparametrized regime that we are talking about. Note that a mildly overparametrized regime would have k = n/d for two-layers and k = \sqrt{n} for 3 layers.
>
> [Theoretical insights into the optimization landscape of over-parameterized shallow neural networks] Thanks for pointing us to this result. This is clearly very relevant and we did not know about this result before, we will definitely discuss the contributions of this paper carefully in our revision. The main idea of this paper is indeed very close to our 2-layer case. However we also note some differences:
> 1. They assume that the samples are generated by a Gaussian distribution, which is a special case under our assumption. Our generalization is very crucial for us to generalize the result to 3-layers, as the output of the first layer is clearly not going to be Gaussian.
> 2. Their result characterizes the optimization landscape (which shows all local minima are globally optimal), but does not actually give an efficient algorithm for finding a local minimum. It is a common mistake to assume that the optimization landscape result directly implies existence of algorithms, however we emphasize that all existing algorithmic guarantees require the function to have smoothness and Hessian-Lipschitzness, which is *NOT* true for the objective as we noted in Section 3.2. We solve this problem by carefully adding a regularizer.
>
> [Questions about 3-layer network result]
> (1) [How to determine p] Actually we do not want the degree p to be larger, we want p to be as small as possible. For the case with no perturbation of the samples, the lower bound on p is the the smallest number such that $x_1^{\otimes 2p},...,x_n^{\otimes 2p}$ is linearly independent (See Theorem 6 for more details). For the case when we add small perturbation on samples, the lower bound should be $p \ge \Omega(\log_d n)$.
> (2) [usefulness of random feature] The random feature layer aims to improve the representation power of the neural network, and this does not help us to change the quadratic activation function. In our 3-layer network, we only train the middle layer and the activation function for hidden layer 2 is also quadratic (See Figure 1 (b) for more details). Our presentation may be a little misleading and we will fix it.
> (3) [fixing top and bottom layer] Yes, this is indeed a restriction to our current techniques. We will emphasize that more in the revision.
> (4) Thanks for your suggestions, we will try our best to reorganize the paper and balance the contents between different sections.
>
> [Proof of Lemma 2] That is the correct intuition, although our current proof is also correct.
>
> [Other Minor comments] Thanks for your suggestions, and we will modify our paper according to your suggestions.

---

### Official Review · AnonReviewer3 · 2019-10-23
**Official Blind Review #3**

**Rating:** 8

**Review:**

The paper studies the amount of over-parameterization needed for a quadratic two (three) layer neural network to memorize a separable training data set with arbitrary labels. The main result of this paper shows that as long as the number of data in dimension d is smaller than d^2/2 and the data is separable by a quadratic function, then a fully connected two-layer neural network with quadratic activation function and 2d hidden neurons can memorize the training data set efficiently.


The intuition behind this result is quite simple: Given data x_1, ..., x_N with a matrix A in R^{d x d} such that x_i^T A^T A x_i = y_i, given the current weight matrix W one can always try to construct an Hessian update direction A' = U A for a column orthonormal matrix U such that x_i^T W^T U A x_i = 0 for every i and (by the property of U) x_i^T A'^TA'x_i = y_i. Note that U has more than d^2/2 degree of flexibility, so having x_i^T W^T U A x_i = 0 is in principle always possible.


The paper studies a very important question: How can neural network memorize data in the mildly over-parameterized regime (number of parameters is linear in the number of data). There is experimental evidence that in this regime, the neural network does memorize much better than its counterpart neural tangent kernel (NTK).



My main concern about this paper is the assumption that the data is separable by a quadratic function, which seems to be very restricted (although this is indeed a progress from those who assume linear separability of the data, e.g.  Brutzkus et al. arxiv:1710.10174,  Wang et al. arXiv:1808.04685 and Oymak & Soltanolkotabi 2019 which has a sub-optimal dependency on the condition number of the training data set). However, in the result such as Allen-Zhu et all, the only requirement is that every data has distance at least \delta, and the final dependency is poly(1/\delta), the assumption is necessary to memorize arbitrary labels over the data set. However, quadratic separable seems to be too strong for a general data set.

Indeed, the authors argued that if the data set is noisy with N < d^2/2, then w.h.p. it is quadratic separable. However, if the network is really using those noise in the data to fit the labels, then one should expect NO GENERALIZATION GUARANTEE of the learnt network at all -- a result that is meaningless for the theory of deep learning.

Missing citation:
Learning Overparameterized Neural Networks via Stochastic Gradient Descent on Structured Data



After Rebuttal: I have read the authors' responses and acknowledge the sensibility of the statement. I agree that the smoothed analysis makes more sense than my original understanding.


**Experience Assessment:**

I have published in this field for several years.

**Review Assessment: Checking Correctness Of Derivations And Theory:**

I carefully checked the derivations and theory.

**Review Assessment: Checking Correctness Of Experiments:**

I did not assess the experiments.

**Review Assessment: Thoroughness In Paper Reading:**

N/A

---

> ### Author Response · Authors · 2019-11-15
> **Response to Reviewer 3**
>
> Dear Reviewer 3,
>
> Thanks for your comments.
>
> [requiring quadratic separable] Although we assume that the 2-tensor of the samples are linearly independent in the 2-layer network structure, we further extend it into the 3-layer network by adding a random feature layer. In this way, we do not require the original data (x) to satisfy the 2-tensor condition. In the 3-layer setting, under the smoothed analysis framework, any dataset with a small perturbation would satisfy our assumption. The quadratic result is exactly what makes our number of parameters tight. If we rely on results such as NTK (which are linear instead of quadratic) then the number of parameters required would be much larger.
>
> We do admit that our assumption is still stronger (or at least harder to interpret) than those which only require the distance between each pair of samples.
>
> [fitting noise] We would like to emphasize that in the smoothed analysis framework, we do not actually perturb the data. Instead, the data is inherently perturbed in the nature. We interpret the smoothed analysis framework as a more quantitative and formal way of saying "as long as the input distribution (x,y) is non-degenerate, our result holds".

---

### Decision · Program_Chairs · 2019-12-19

**Decision:**

Reject

**Comment:**

The paper studies the amount of over-parameterization needed for a quadratic 2 /3 layer neural network to memorize a separable training data set with arbitrary labels. While the reviewers agree that this paper contains interesting results, the review process uncovered highly related prior work, which requires a major revision to put the current paper into perspective and generally various clarifications. The paper will benefit from a revision and resubmission to another venue, and is in its current form not ready for acceptance at ICLR-2020.